# A genetic, genomic, and computational resource for exploring neural circuit function

**Fred P Davis[1,2†‡*], Aljoscha Nern[1†], Serge Picard[1], Michael B Reiser[1], Gerald M Rubin[1], Sean R Eddy[1,3,4], Gilbert L Henry[1,5*]**

[1]Janelia Research Campus, Howard Hughes Medical Institute, Ashburn, United States; [2]Molecular Immunology and Inflammation Branch, National Institute of Arthritis and Musculoskeletal and Skin Diseases, National Institutes of Health, Bethesda, United States; [3]Howard Hughes Medical Institute and Department of Molecular and Cellular Biology, Harvard University, Cambridge, United States; [4]John A. Paulson School of Engineering and Applied Sciences, Harvard University, Cambridge, United States; [5]Cold Spring Harbor Laboratory, Cold Spring Harbor, United States

**\*For correspondence:**
fredpdavis@gmail.com (FPD);
henry@cshl.edu (GLH)

[†]These authors contributed equally to this work

**Present address:** [‡]Celsius Therapeutics, Cambridge, United States

**Competing interests:** The authors declare that no competing interests exist.

**Abstract** The anatomy of many neural circuits is being characterized with increasing resolution, but their molecular properties remain mostly unknown. Here, we characterize gene expression patterns in distinct neural cell types of the *Drosophila* visual system using genetic lines to access individual cell types, the TAPIN-seq method to measure their transcriptomes, and a probabilistic method to interpret these measurements. We used these tools to build a resource of high-resolution transcriptomes for 100 driver lines covering 67 cell types, available at http://www.opticlobe.com. Combining these transcriptomes with recently reported connectomes helps characterize how information is transmitted and processed across a range of scales, from individual synapses to circuit pathways. We describe examples that include identifying neurotransmitters, including cases of apparent co-release, generating functional hypotheses based on receptor expression, as well as identifying strong commonalities between different cell types.

## Introduction

The anatomy of neural circuits is being characterized with increasing resolution and throughput, in part following a dramatic increase in the size of circuits amenable to detailed electron microscopy reconstruction (*Swanson and Lichtman, 2016*) and the development of genetic tools to access individual cell types (*Luo et al., 2018*). These efforts reveal anatomy at unprecedented detail, but not the molecular properties of cells. In principle, the genes expressed in each cell of a neural circuit should serve as a molecular proxy for cell physiology. However, most genomic efforts have focused on surveying neuronal diversity rather than characterizing circuit function (*Ecker et al., 2017*). To develop a resource exploring molecular correlates of circuit function, here we use an approach that genetically targets cell types within a well-characterized brain region to measure high-quality transcriptomes that can be integrated with connectomes.

*Drosophila* affords an ideal system to study neural circuits in detail, as both excellent genetic tools and high resolution connectomes are available. Here we focus on the repeating columnar circuits of the visual system, found in the optic lobes, a widely used model for studying circuit development and function with an extensive genetic toolbox and well-described anatomy (*Figure 1A*; *Nériec and Desplan, 2016*; *Silies et al., 2014*; *Apitz and Salecker, 2014*). This network begins with photoreceptor neurons and contains several layers of connected neurons which process incoming

**eLife digest** In the brain, large numbers of different types of neurons connect with each other to form complex networks. In recent years, researchers have made great progress in mapping all the connections between these cells, creating 'wiring diagrams' known as connectomes.

However, charting the connections between neurons does not give all the answers as to how the brain works; for example, it does not necessarily reveal the nature of the information two connected cells exchange. Assessing which genes are switched on in different neurons can give insight into neuronal properties that are not obvious from physical connections alone.

To fill that knowledge gap, Davis, Nern et al. aimed to measure the genes expressed in a well-characterized network of neurons in the fruit fly visual system. First, 100 fly strains were established, each carrying a single type of neuron colored with a fluorescent marker. Then, a biochemical approach was developed to extract the part of the cell that contains the genetic code from the neurons with the marker. Finally, a statistical tool was used to assess which genes were on in each type of neurons. This led to the creation of a database that shows whether 15,000 genes in each neuron type across 100 fly strains were switched on.

Combining this information with previous knowledge about the flies' visual system revealed new information: for example, it helped to understand which chemicals the neurons use to communicate, and whether certain cells activate or inhibit each other.

The work by Davis, Nern et al. demonstrates how genetic approaches can complement other methods, and it offers a new tool for other scientists to use in their work. With more advanced genetic methods, it may one day become possible to better grasp how complex brains in other organisms are organized, and how they are disrupted in disease.

luminance signals into multiple parallel streams of visual information (*Figure 1B*). Many of its cellular components have been described by light microscopy, including classical Golgi studies (*Fischbach and Dittrich, 1989*) and recent analyses using genetic methods (*Morante and Desplan, 2008*; *Otsuna and Ito, 2006*; *Nern et al., 2015*; *Wu et al., 2016*). Electron microscopy reconstruction work has characterized the synaptic connections of many optic lobe neurons (*Meinertzhagen and O'Neil, 1991*; *Meinertzhagen and Sorra, 2001*; *Rivera-Alba et al., 2011*; *Takemura et al., 2013*; *Takemura et al., 2015*; *Takemura et al., 2017*; *Shinomiya et al., 2019*). Comparative studies have also explored the evolution of this ancient brain structure (*Strausfeld, 2009*). Despite this wealth of information, many of its fundamental properties remain unknown, including the neurotransmitters used at many of its synapses.

Measuring the genes expressed in specific cells of the brain is challenging due to its compact and complex organization. RNA sequencing (RNA-seq) addresses this challenge by profiling either single cells or genetically labeled populations of cells (*Ecker et al., 2017*). The latter approach requires genetic tools to access individual cell types but provides more direct access to cells of interests than sampling of unmarked single cells, especially for sparse cell types. Profiling identified cell types provides a direct link to previous work on the anatomy and physiology of those cell types. Cell type-specific drivers also facilitate follow-up experiments, for example evaluating the role of individual genes in individual cells. In *Drosophila*, large collections of GAL4 driver lines (*Jenett et al., 2012*; *Tirian and Dickson, 2017*) and the possibility to further refine these patterns with intersectional methods such as split-GAL4 (*Luan et al., 2006*; *Dionne et al., 2018*) enable genetic access to many neuronal populations (see, for example, *Tuthill et al., 2013*; *Aso et al., 2014*; *Wu et al., 2016*). We therefore chose the genetic, rather than single cell, approach to build a genomics resource to explore circuit function.

We previously developed an Isolation of Nuclei Tagged in a specific Cell Type (INTACT) method (*Deal and Henikoff, 2010*) to measure transcriptomes and epigenomes of genetically-marked neuronal populations in *Drosophila* (*Henry et al., 2012*) and mouse (*Mo et al., 2015*). Here, we develop a tandem affinity purification of INTACT nuclei (TAPIN) method with increased specificity, sensitivity, and throughput. By combining this method with an extensive set of new driver lines with predominant expression in specific cell types and a new probabilistic method to interpret transcript abundance, we build a resource of high-quality transcriptomes for one hundred driver lines. We selected

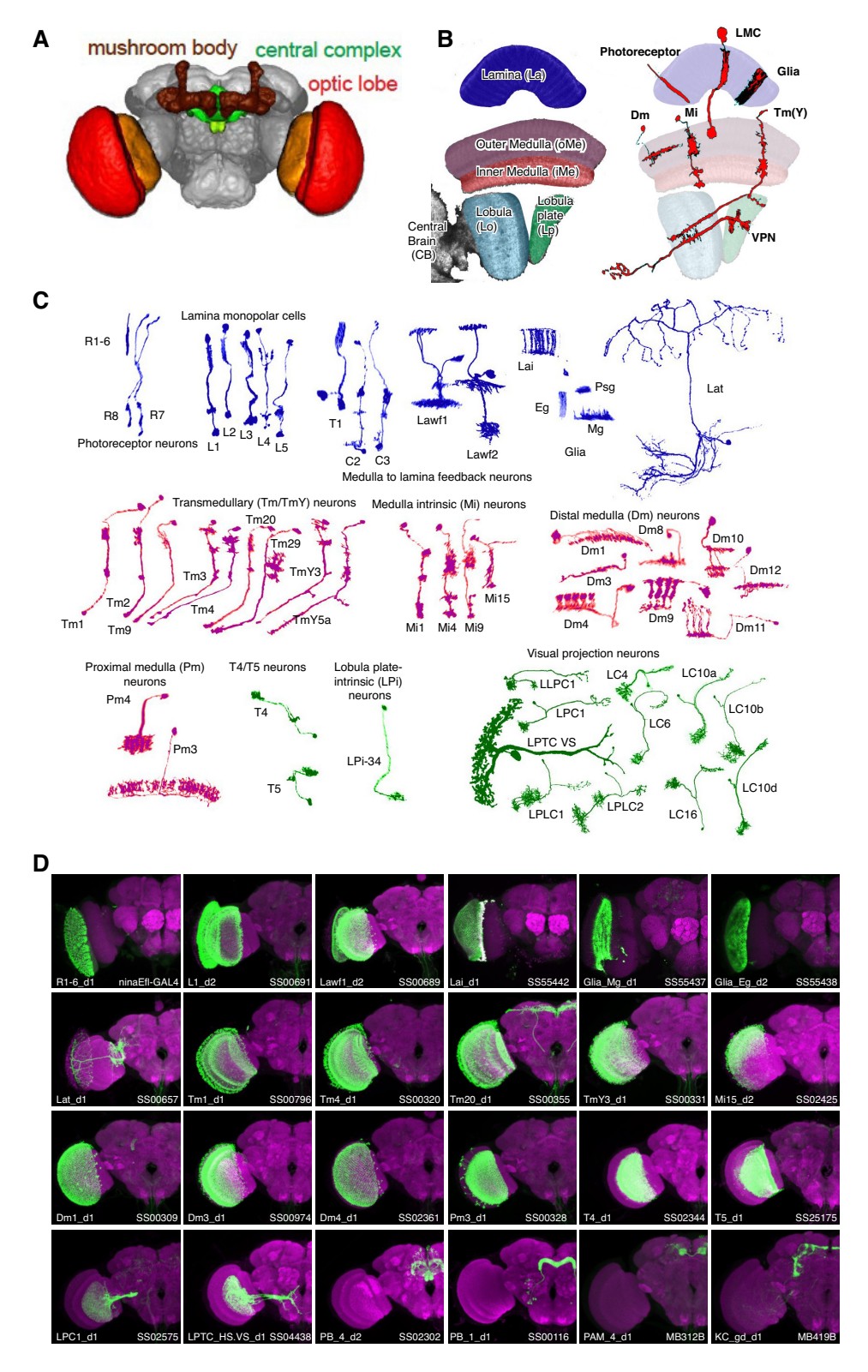

**Figure 1.** Genetic tools to access cell types in the *Drosophila* visual system. (**A**) Major brain regions profiled in this study (brain image from *Jenett et al., 2012*). The optic lobes have a repetitive structure of ~750 retinotopically arranged visual columns of similar cellular composition. (**B,C**) Examples of single cells in the optic lobe. (**B**) Left, subregions of the fly visual system. Right, examples of layers and neuropil patterns of various classes of visual system neurons. (**C**) We profiled cell types arborizing in the lamina (blue), medulla (purple) and lobula complex (green) of the visual system. *Figure 1 continued on next page*

*Figure 1 continued*

Many cells contribute to multiple neuropiles so other groupings are possible. Note, some cell types are present at one cell per column, while others are less numerous with cells that each contribute to several columns. For example, the main synaptic region of the first optic lobe layer, the lamina, contains processes of some 13,000 cells but these belong to only 17 main cell types: 14 neuronal and three glial (*Figure 1C*, top row). A small number of additional neurons (lamina tangential cells, Lat) project to a region just distal to the main lamina neuropile. (D) Representative expression patterns of driver lines that target specific cell types. Each image is a maximum intensity projection of a whole brain confocal stack (only one optic lobe is shown). In each image the brain is counter-stained (magenta) with a neuropil marker and both the targeted cell type and the driver are indicated in the lower left and right corner, respectively. Additional images (focusing on drivers first described in this study) are shown in *Figure 1—figure supplements 1* and *2*. Imaging parameters and brightness and contrast were adjusted individually for each image. For genotypes and image details see *Supplementary file 1E*.

The online version of this article includes the following figure supplement(s) for figure 1:

**Figure supplement 1.** Whole brain expression patterns of new driver lines generated in this study.
**Figure supplement 2.** Optic lobe patterns of driver lines.

---

drivers that expressed in cell types constituting the lamina (*Fischbach and Dittrich, 1989*; *Tuthill et al., 2013*; *Edwards et al., 2012*) as well as the major cell types of the circuits that compute the direction of visual motion (*Mauss et al., 2017*; *Figure 1C*). We further included neuronal populations in two central brain regions, the mushroom body and central complex, primarily to serve as informative outgroups.

By profiling these driver lines, we develop an expression catalog for 67 *Drosophila* cell types as well as several broader cell populations. Through validation experiments and comparisons to the literature we demonstrate that this resource is useful both for identifying individual genes expressed in specific cell types and for revealing broader patterns such as the expression of all members of a gene family across many cell types. As an example, we describe the expression of neurotransmitters and their receptors and use this information to interpret synaptic connectivity. For example, we unexpectedly found that the R8 photoreceptors express acetylcholine in addition to histamine and show that this apparent co-transmitter phenotype is further supported by differential expression of neurotransmitter receptors in R8 postsynaptic partners. Our results demonstrate that combining expression and connectomes leads to specific testable hypotheses about circuit mechanisms that are inaccessible to either approach alone.

## Results

### Genetic tools for labeling the visual system

To enable transcriptome analyses of defined cell populations, we first assembled a collection of genetic drivers to access them. For this study, we combined drivers from existing collections for cell types in the lamina (*Tuthill et al., 2013*), the mushroom body (*Aso et al., 2014*), and the lobula (*Wu et al., 2016*) with new driver lines for many additional optic lobe cell types and also some neurons of the central complex (*Wolff and Rubin, 2018*), T. Wolff, personal communication). Nearly all of these drivers were generated using an intersectional method, split-GAL4, to refine expression patterns of GAL4 driver lines. To characterize new driver lines, we imaged expression patterns across the entire fly brain to determine overall driver specificity (*Figure 1D*, *Figure 1—figure supplement 1*) and examined anatomical features such as layer patterns in higher resolution images to identify specific cell types (*Supplementary file 1A*, *Figure 1—figure supplement 2*). For most lines, we further confirmed the identity of labeled cells by examining the morphology of individual cells using stochastic labeling (*Figure 1—figure supplement 2*). Although we noted that a few patterns also include some additional contaminating cells (*Supplementary file 1A*), these driver lines are the most specific tools currently available to access individual cell types in the optic lobe.

### Purifying nuclei with TAPIN

Next, we employed an improved INTACT method to measure nuclear transcriptomes in genetically defined cell populations (*Henry et al., 2012*), and we also developed a new variant of the method that permits higher throughput with increased purity and sensitivity. In both approaches, nuclei are purified using a nuclear tag whose expression is driven in a cell population of interest by either a

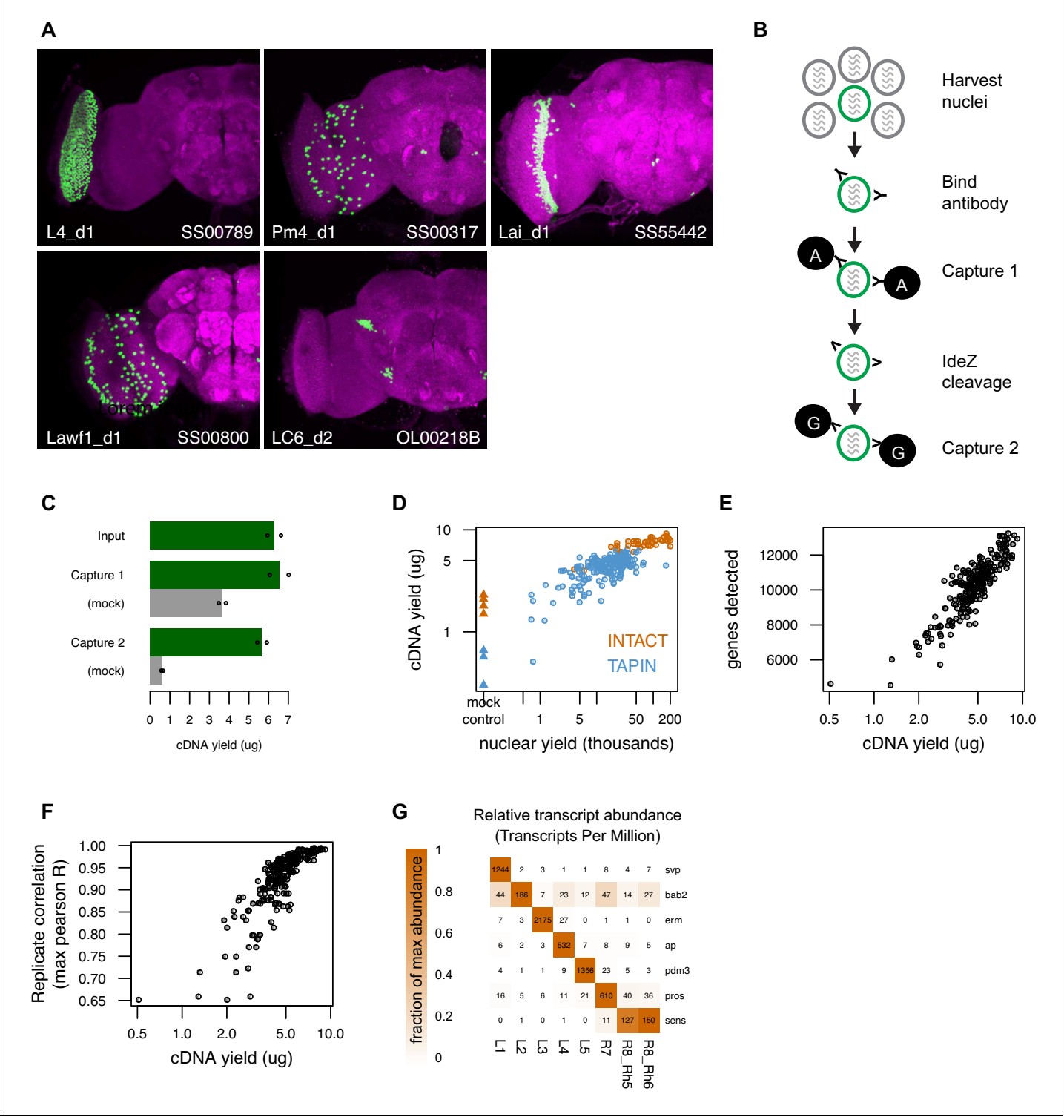

**Figure 2.** Tandem-affinity purification of INTACT nuclei (TAPIN) enables neuronal genomics. (**A**) Cell type-specific drivers enable expression of the UNC84-2XGFP nuclear tag (green) in specific populations of cells. Both the targeted cell type and driver are indicated in the lower left and right corner, respectively. (**B**) Following nuclei harvest, two rounds of magnetic bead capture serially purify target nuclei. After the first round of protein A bead capture, bacterial protease IdeZ cleaves the anti-GFP antibody in the flexible hinge region, allowing a second round of bead capture with protein G, which recognizes the F(ab')$_2$ region. Protein G, unlike Protein A, can bind both the Fc and F(ab')$_2$ regions of an immunoglobulin. (**C**) Two capture rounds reduce the level of non-specific background (gray bars, mock IgG control) while maintaining the cDNA yield from the captured target nuclei (green bars). Bars represent the mean of two replicates (shown as points). (**D**) RNA-seq libraries created with more nuclei yield more cDNA (circles).
*Figure 2 continued on next page*

*Figure 2 continued*

TAPIN libraries had lower non-specific background than INTACT (blue *vs* orange triangles). (E) Libraries with more cDNA detect more genes. (F) Libraries with more cDNA have more reproducible transcript abundances. (G) Previously identified markers of lamina monopolar and inner photoreceptor neurons (*Tan et al., 2015*) are enriched in the expected cells.

The online version of this article includes the following figure supplement(s) for figure 2:

**Figure supplement 1.** Two variants of nuclei capture, INTACT and TAPIN.
**Figure supplement 2.** TAPIN-seq vs FACS-seq comparison.

standard or split GAL4 driver (*Figure 2A*). The new variant protocol, t̲andem a̲ffinity p̲urification of I̲NTACT n̲uclei (TAPIN), uses a bacterial protease (IdeZ) to specifically cleave antibodies in the hinge region separating their Fc and antigen binding F(ab')$_2$ fragments (*Figure 2B*, *Figure 2—figure supplement 1B*). Treating protein A magnetic bead-bound nuclei with this protease generates both nucleus-F(ab')$_2$ and bead-Fc complexes. Soluble nucleus-F(ab')$_2$ is then recaptured on protein G magnetic beads, removing non-specifically bound material from the first capture. INTACT successfully profiled many of the abundant cell types in the optic lobe (>1000 cells per brain), but failed for sparser cell types and those whose nuclei were difficult to purify by differential centrifugation (photoreceptors, glia, T4, T5). We solved these problems with TAPIN, which does not purify nuclei prior to bead capture.

The greatest advantage of TAPIN is its ability to purify nuclei from sparse cell types (<50 cells/brain) (*Supplementary file 1A*). INTACT is not suitable for these lines because of loss during differential centrifugation. This difficulty cannot be overcome by processing more brains per experiment because differential centrifugation is difficult to scale. TAPIN solves this problem by running a first capture on crude extracts generated from hundreds to thousands of fly heads. The substantial background in this first capture is reduced 5- to 6- fold in a second capture with only a modest decline in both the yield of nuclei and amplified cDNA (*Figure 2C*).

## Measuring transcriptomes with INTACT- and TAPIN-seq

We applied INTACT and TAPIN to the cell populations defined by the genetic drivers we described above (*Supplementary file 1B*). Most drivers express in a single anatomically defined cell type or a small group of related cell types. We note that a few of our cell types are strictly groups of related cell types (for example, the muscle cells or, at a different level of a cell type hierarchy, the T4 and T5 cells, with four subtypes each). Additional drivers target more heterogeneous cell populations sharing a common property (e.g., driver lines aimed at recapitulating the expression of a neurotransmitter marker). Altogether, we built 250 RNA-seq libraries from 242 samples of purified nuclei (46 using INTACT and 196 using TAPIN) and eight manually dissected samples (*Supplementary file 1B*). We estimated relative transcript abundance in each library using kallisto (*Bray et al., 2016*). Libraries built from more nuclei yielded more cDNA (*Figure 2D*), allowed more genes to be detected (*Figure 2E*), had more estimated transcripts (*Figure 2—figure supplement 1C*), more reproducible transcript abundance (*Figure 2F*), and less bias in coverage across gene bodies (*Figure 2—figure supplement 1D,E*). We focused on 203 libraries that had at least 8500 genes detected, 3µg cDNA yield, and 0.85 Pearson's correlation of transcript abundances in two biological replicates. These 203 libraries consist of at least two biological replicates built from 100 drivers that covered 67 cell types (53 visual system, 7 mushroom body, 5 central complex, 2 muscle), 6 broader cell populations (ChAT, Gad1, VGlut, Kdm2, Crz, and NPF), and 2 manually dissected tissues (the lamina and remainder of the optic lobe) (Materials and methods). We provide the read and abundance data for the remaining sub-optimal libraries (47 libraries covering 24 cell types) in the event they may be informative, but we do not consider these to be of sufficient quality and do not consider them further here. We did not sort the sex of flies when preparing TAPIN-seq libraries, as we did not observe large differences in male and female expression profiles (*Figure 2—figure supplement 1F*).

We were encouraged by the clear enrichment of previously identified markers in cell types where they were expected. For example, we recovered transcription factors (TFs) previously found in the developing monopolar interneurons and inner photoreceptors (*Tan et al., 2015*; *Figure 2G*). We further confirmed our measurements by comparing TAPIN-seq results for twelve cell types that were also recently profiled by FACS-seq (*Konstantinides et al., 2018*; *Figure 2—figure supplement 2A,*

*B*) and found concordant expression of cell type-enriched genes. This concordance also argues against major differences between nuclear and cytoplasmic transcriptomes. In combination with the technical quality of our libraries, this confirmation by independent gene expression measurements validated our approach, and also motivated us to explore how to best interpret a large dataset of relative abundances.

## Interpreting transcript abundance with mixture modeling

Deriving biological insights from a matrix of transcript abundances is not straightforward. While a cell's expression of a gene can be used to infer a specific functional property of that cell, the level of expression that is needed to establish confidence in such an inference is much less clear. For example, expressing the vesicular acetylcholine transporter (*VAChT*) implies that a neuron is cholinergic. However, *VAChT* transcript abundance exhibits a wide distribution and it is not clear, a priori, what level is necessary to conclude that a cell is cholinergic (*Figure 3A*).

We used mixture modeling to address this challenge by describing the expression levels of each gene as arising from a mixture of two log-normal distributions representing binary 'on' and 'off' states (*Figure 3A*; Materials and methods). Genes can of course express in more than two states, but we show through extensive validation that this simplifying assumption is a useful one. Modelling *VAChT* expression in the high-quality TAPIN/INTACT-seq libraries unambiguously inferred *VAChT* states for all drivers (*Figure 3B*). We also found that the model was useful for addressing transcript-carryover, evident in our data (as well as published bulk and single cell studies in the fly [*Davie et al., 2018*] and mouse [*Siegert et al., 2012*; *Macosko et al., 2015*]) as photoreceptor transcripts detected in non-photoreceptor cells (*Figure 3—figure supplement 1A*). For example, the model correctly inferred that only R1-6 photoreceptors expressed the primary rhodopsin *ninaE*, although *ninaE* abundance in other cells reached as high as 2,702 TPM (the mushroom body cell type PAM_1) (*Figure 3—figure supplement 1B,C*). We used this method to transform our catalog of transcript abundances to probabilities of expression (*Figure 3C*), observing a wide spectrum of on levels and dynamic ranges between on and off states (*Figure 3—figure supplement 1D,E*). To further simplify these probabilities, we discretized them into on ($p>=0.8$) and off ($p<=0.2$) states, and otherwise considered them to be ambiguous ($0.2 < p < 0.8$). The expression states inferred for replicates had a median 95% concordance (*Figure 3—figure supplement 1F*). We combined information from replicates to infer expression at the driver and cell type levels (Materials and methods).

We found many genes that express in all cell types, and many that express in only one, with a range in between (*Figure 3D,E*). As expected, given their roles in specifying identity, homeobox transcription factors (TF) expressed more specifically than transcription factors in general (*Figure 3E*). Neuropeptides also expressed specifically, while genes with the more general function of synaptic vesicle endocytosis were broadly expressed. We explore these functional properties in more detail later (*Figure 4C*).

## Evaluating accuracy of TAPIN-seq measurements

To validate our TAPIN-seq measurements, we first compared our inferred expression states to Fly-Base curated reports of protein expression (n = 197 data points of gene/cell pairs; four negative points, 193 positive points; n = 22 cells; n = 69 genes, *Supplementary file 1C*). Protein expression can of course differ from that of the corresponding mRNA due to post-transcriptional regulation. However, since most functional interpretations of transcriptome data are implicitly about protein expression, we used this as a more stringent and practical test of our model. We found 93% concordance (183 matches; 14 mismatches from six genes; 0 mismatches for negative benchmark points; *Figure 3—figure supplement 1G*). The benchmark mismatches fell into three categories: expression levels near the transition between inferred on and off components (*veli*, *verm*, *para*; *Figure 3—figure supplement 1H–J*), genes with a wide dynamic range of expression (*Syx*, *Rab11*; *Figure 3—figure supplement 1K,L*), and genes with undetected transcript but previously detected protein (*Myo61F*; *Figure 3—figure supplement 1M*). The first two categories likely arise from imprecision in the model's fitted components and its representation of transcript abundance as bimodal, rather than continuous. The third category (conflicting transcript and protein levels) could reflect either technical issues (low sensitivity in our measurements, or false positives in the prior work due to

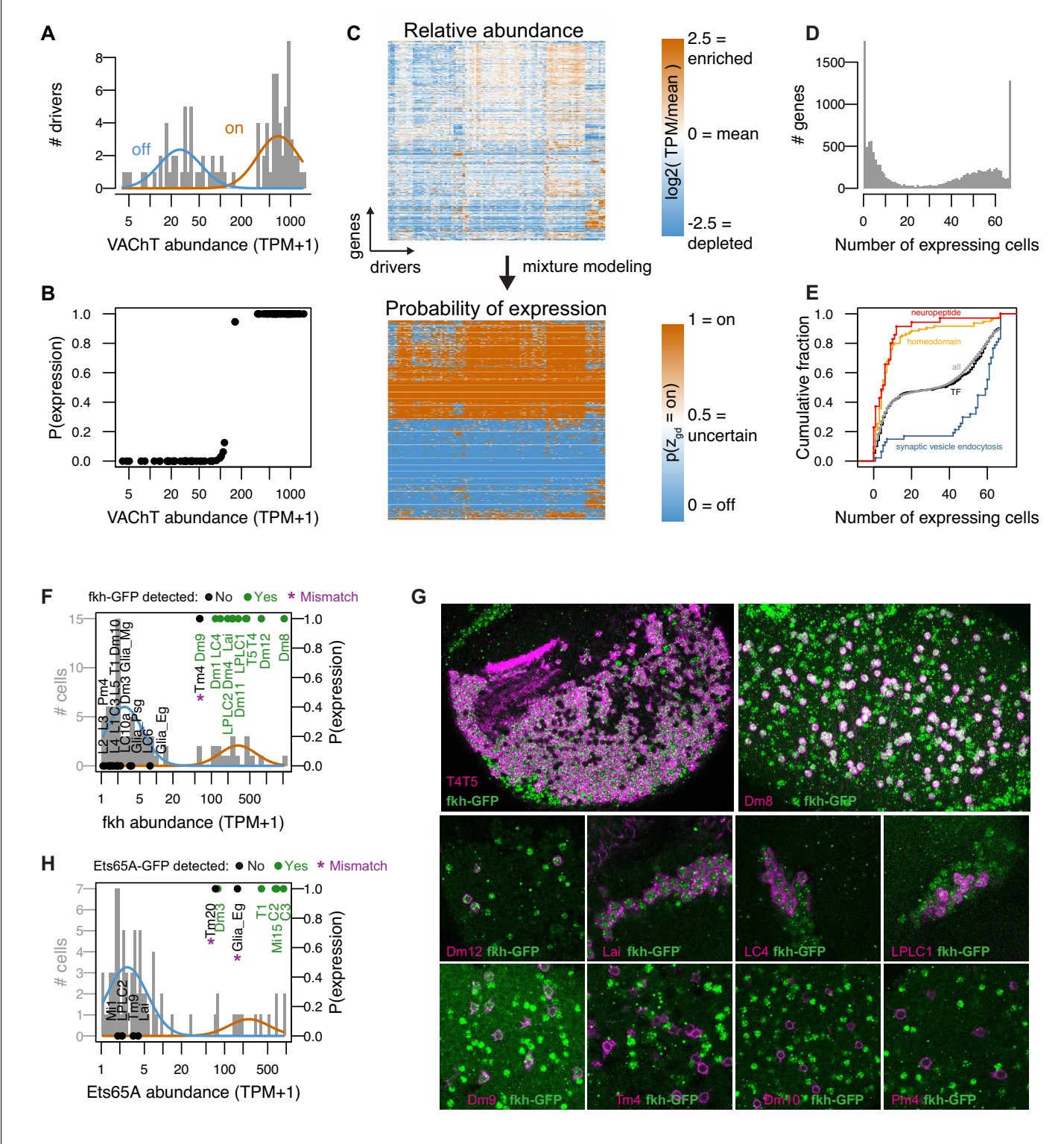

**Figure 3.** Mixture modeling accurately interprets TAPIN-seq measurements. (A) The distribution of Vesicular acetylcholine transporter (*VAChT*) abundance fit with a mixture of two log-normal components. (B) Interpreting these components as 'off' and 'on' states unambiguously infers expression state in essentially all drivers. (C) Mixture modeling transforms our catalog of relative transcript abundances (top) to discretized expression states (bottom). (D) Histogram of expression breadth per gene. (E) Cumulative distributions of expression breadth for all genes (gray), transcription factors (black), homeobox TFs (orange; InterPro domain IPR001356), neuropeptides (red), and genes involved in synaptic vesicle endocytosis (blue). (F,G) The *fkh* modeling results were compared to its protein expression pattern as evaluated with a BAC transgenic (See *Figure 3—figure supplement 2A*). (F)

*Figure 3 continued on next page*

*Figure 3 continued*

Histogram bars represent raw abundance of all cells in our catalog. Blue and orange curves represent the inferred off and on components, respectively. Points represent the cells tested for transgene expression showing either detectable GFP (Green) or no signal (Black). The points' vertical position reflect the estimated probability of gene expression. (G) Forkhead-GFP expression in selected cell types. Fkh-GFP (mainly nuclear, in green) and cell type-specific expression of a membrane marker (in magenta) are shown. Because of the wide range of fkh expression levels, imaging parameters and brightness and contrast adjustments are not identical for different panels. Cells with detectable nuclear GFP signal above the background in the same image were scored as expressing fkh. (H) As in J, to evaluate Ets65A modeling results (See *Figure 3—figure supplement 2B*).

The online version of this article includes the following figure supplement(s) for figure 3:

**Figure supplement 1.** Overview of INTACT-seq and TAPIN-seq libraries.
**Figure supplement 2.** Validation of fkh and Ets65A model inferences.

antibody cross-reaction) or biological complexities (e.g., long-lived transcripts, subcellular localization, post-transcriptional regulation).

To further evaluate our results for genes expressed across a wide range of levels, we compared the model output to protein expression patterns for two transcription factors: Forkhead (*fkh*) and *Ets65A*. We visualized each protein using a C-terminal GFP tag; the tagged proteins were expressed from BAC transgenes with large flanking sequences to ensure a near native genomic context (*Kudron et al., 2018*). From the transcript data, we inferred *fkh* gene expression in 14 cell types across a 35-fold range of abundance (60 to 2,103 TPM). Of 28 cell types that we visualized at the protein level, fkh was detected in all but one that we expected from TAPIN-seq (*Figure 3F,G*, *Figure 3—figure supplement 2A*). The sole exception, Tm4, has a *fkh* abundance (60 TPM) near the border between the inferred off and on states (*Figure 3F*). However, we did detect protein in Dm9, which had a near identical raw transcript abundance (61 TPM). Similarly evaluating *Ets65A* expression identified two mismatches out of 11 tested cells (*Figure 3H*, *Figure 3—figure supplement 2B*). Ets65a protein was not detected in Tm20 (70 TPM) and epithelial glia (161 TPM), while it was weakly detected in Dm3 (77 TPM). These results further support the accuracy of TAPIN-seq and our statistical model even for genes with a wide dynamic range. The agreement between our transcript on/off calls and protein expression encouraged us to use the discretized on/off calls for all further analyses; the unprocessed relative abundances in TPM are reserved for deeper analysis when needed.

## Examining the relation between cell types using transcriptomes

To study the relation between cell types, we built a dendrogram based on the expression states we inferred for the whole transcriptome and estimated the support for each branch point with bootstrap resampling (*Figure 4A*). The broad groupings were well supported and mostly intuitive: muscle were outgroups, followed by a mushroom body cell type (PAM_4), the glia, the photoreceptors, and the remaining neurons. Several fine groupings of anatomically closely related neurons were also well supported (e.g., Kenyon cells; C2, C3; Lawf1, Lawf2; T4, T5; LPLC1, LPLC2). However, mid-level branchings were not well supported, indicating the lack of a simple hierarchical relationship. Neurons were generally grouped by region: central complex, mushroom body, and optic lobe. One surprise was the grouping of Tm20 and Dm1, away from all other optic lobe cell types. Upon closer examination, the identity of genes expressed exclusively in these two lines (*lz*, *Pdh*, *bw*) suggest that this grouping is driven by shared pigment cell contamination in the GAL4-tagged patterns of these driver lines. Similarly, the unusual position of PAM_4 is likely due to some unidentified non-neuronal cells in the driver. These are examples of imperfections in the GAL4 driver lines. While they can lead to some false positives for the main target cell types, they can also provide additional information. For example, analyzing the overlap between Dm1 and Tm20 allowed us to infer marker genes expressed in the pigment cell population.

## Identifying genes that mark cell types and groups

We next identified genes that marked cell groups in the tree, using three criteria: genes that expressed in all the cells within a group, at most two cells outside this group, and with transcript abundance higher than all cells outside the group (For simplicity, we will hereon refer to cell type as just cell. This is not meant to exclude the possibility of heterogeneity within the individual cells of a profiled population.). We used these criteria to identify markers for photoreceptors (n = 108), glia

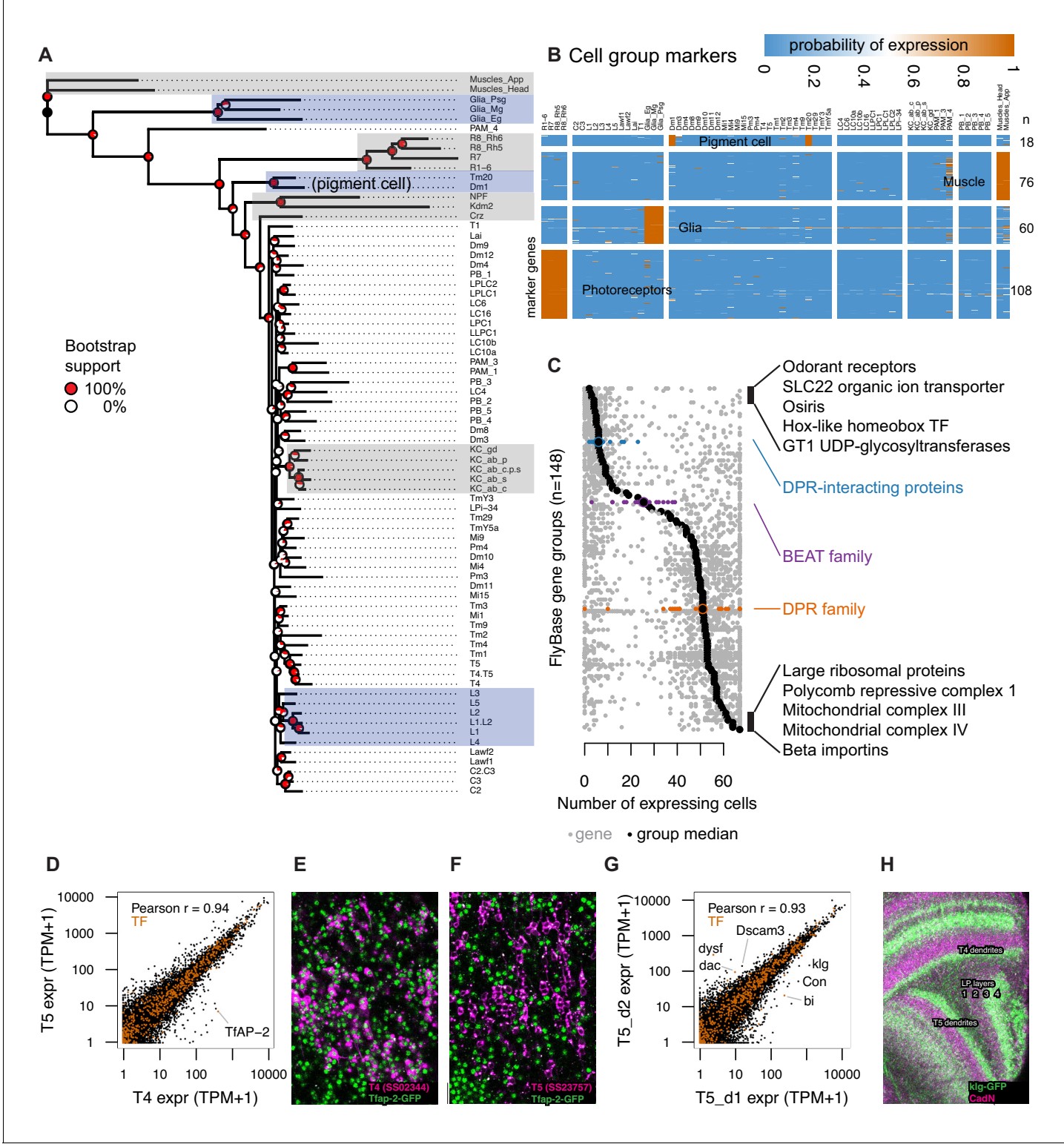

**Figure 4.** TAPIN-seq profiles identify genes enriched in cell types and groups. (**A**) Cells grouped by a minimum evolution tree of their inferred expression states. (**B**) Heatmap of marker genes enriched in photoreceptors, glia, muscle, and pigment cells. (**C**) Distribution of expression breadth for genes in 'terminal' FlyBase gene groups with more than 10 members in our expression probability matrix. The least- and most- broadly expressed gene groups are labeled, along with the DPR-interacting, beat and DPR family of extracellular proteins. (**D**) TfAP-2 transcription factor distinguishes closely related cell types T4 and T5. (**E,F**) TfAP-2 protein is specifically expressed in T4 and not in T5, confirming this detection of differential expression levels. GFP-tagged TfAP-2 (mainly nuclear, in green; see *Supplementary file 1E* and Materials and methods) is shown together with a membrane marker

*Figure 4 continued on next page*

*Figure 4 continued*

(magenta) expressed in T4 (**E**) or T5 (**F**) cells. (**G**) Comparison of genes with differential expression in two driver lines for T5 neurons expressing in different subtypes, identify genes that differentially label layers of the lobula plate (corresponding to different subtypes of T5 cells). (**H**) Confirming the TAPIN-seq identification, klg protein (detected using a GFP tag (green); see ***Supplementary file 1E*** and Materials and methods) is expressed in T4/T5 cells with the expected layer specificity (layers 3 and 4) in the lobula plate (LP). A neuropil marker is shown in magenta.

The online version of this article includes the following figure supplement(s) for figure 4:

**Figure supplement 1.** TAPIN-seq profiles identify genes enriched in cell types and groups.

(n = 60), and muscle (n = 76) (***Figure 4B***, ***Supplementary file 1D***). These genes included many known as well as new markers. For example, genes enriched in photoreceptors include signaling components (*Arr2*, *Galphaq*) and transporters (*trpl*, *Eaat2*) with known physiological roles as well as uncharacterized orphan transporters (e.g., *CG8468*). We also identified 18 markers for pigment cells using the Tm20 and Dm1 profiles. In addition to the three types of lamina glia we profiled, several other glia types are present in both the lamina and the medulla. Genes expressed exclusively in the dissected samples (lamina, remainder of optic lobe) and not in the TAPIN libraries identified marker genes for optic lobe cells that we did not directly profile, such as medulla glia. Indeed, the genes identified in this way included several known markers for astrocytes (*alrm*, *wun2*, *Obp44a*) (***Huang et al., 2015***).

We examined the breadth of expression of different functional groups of genes, as defined by FlyBase gene group curation. HOX-like homeobox TFs were among the most specifically expressed group, while groups of core cellular machinery (e.g., beta importins, mitochondrial complexes) were among the most broadly expressed groups (***Figure 4C***). Some groups included both broadly and very specifically expressed genes. For example, among cell adhesion molecules, we noted an interesting distribution for three gene groups proposed to be involved in protein-protein interactions that underlie synaptic connectivity (***Özkan et al., 2013***; ***Tan et al., 2015***; ***Carrillo et al., 2015***). While 11 DPR-interacting proteins (DIP) were among the most specifically expressed genes (expressed in a median of 6 cells), beat (median, 25.5 cells) and DPR (median, 51 cells) genes were more broadly expressed (***Figure 4—figure supplement 1A–D***). As physical interactions among these and other extracellular proteins have been systematically characterized (***Özkan et al., 2013***), we combined their expression and interaction patterns to estimate the number of potential interaction between cells in the lamina (***Figure 4—figure supplement 1E***), many of which are in actual contact (***Figure 4—figure supplement 1F***). Except for a clear paucity of transcripts encoding interacting protein pairs expressed by glia, these global expression-based patterns did not correlate well with connectivity in the lamina. However, we found that every pair of lamina cells expressed mRNAs for tens of interacting protein pairs, highlighting the broad potential for cell-cell interactions not only in the developing (***Tan et al., 2015***) but also adult optic lobe.

## Transcriptomes can distinguish closely related cell types and subtypes

To ask whether we could identify genes distinguishing closely related cell types, we examined T4 and T5. These cells had similar transcriptomes and were neighbors in the phylogenetic tree, but we found one transcription factor, *TfAP-2*, that was expressed nearly two orders of magnitude higher in T4 (390 TPM) than T5 (6 TPM) (***Figure 4D***). We confirmed this pattern at the protein level (***Figure 4E,F***).

T4 and T5 cells can each be further divided into four subtypes that preferentially respond to motion in one of four cardinal directions and differ in anatomical details such as the lobula plate layer to which they project axons. While our split-GAL4 lines do not isolate single T4/T5 subtypes, the T5_d1 and T5_d2 drivers show differences in subtype expression (***Figure 1—figure supplement 1B,B',C,C'***). Comparing the transcriptomes of these two drivers confirmed previously described markers (*Con*, *bi*, *dac*; ***Apitz and Salecker, 2018***) that distinguish T4/T5 cells of lobula plate layers 1/2 and 3/4, and indicated additional genes, including a transcription factor (*dysf*) and cell adhesion molecules (*klg*, *Dscam3*) with selective expression in these subtypes (***Figure 4G***). As a further confirmation of this finding, we verified that a tagged klg protein showed layer-specific expression in the lobula-plate consistent with these T4/T5 subtypes (***Figure 4H***).

## Reference bulk transcriptomes help interpret single cell transcriptomes

Single cell RNA-seq (scRNA-seq) was recently used to map the optic lobe (*Konstantinides et al., 2018*) and brain (*Davie et al., 2018*). Despite its routine use, interpreting scRNA-seq measurements – clustering single cell transcriptomes and labeling these clusters as known cell types – remains challenging. For example, the 52 single cell clusters found in the optic lobe (23 labeled as known cell types, including 7 types of glia; *Konstantinides et al., 2018*) and the 87 clusters found in the whole brain (41 labeled, also including seven glia; *Davie et al., 2018*) far under-estimate the expected diversity of cell types – over one hundred anatomically distinct neuronal cell types have been described in the optic lobe alone (*Fischbach and Dittrich, 1989*; *Morante and Desplan, 2008*; *Otsuna and Ito, 2006*; *Nern et al., 2015*; *Wu et al., 2016*). Furthermore, comparing the number of single cells in each optic lobe cluster to the true abundance of each cell type (as established by neuroanatomical studies) reveals that the single cell map does not proportionally represent abundance (ranging from ~5 times fewer Dm8/Tm5c cells to ~7 times more Dm12 cells than expected in the optic lobe map; *Figure 5A*). The whole brain map, measured using the more sensitive 10X scRNA-seq platform rather than Drop-seq used for the optic lobe map (*Svensson et al., 2017*), showed similar cell type abundances (*Figure 5B*). Without detailed neuroanatomy to serve as ground-truth, this similarity could be interpreted as reproducibility across platforms. Instead, our results suggest caution when interpreting cell type frequencies from scRNA-seq maps, as they can be skewed by experimental artifacts such as cell type-specific differences in RNA isolation yields, computational over-clustering, or inaccurate cell type labeling. Given the known number of cell types and their frequencies, it is clear that interpreting single cell measurements is challenging.

Comparison with our data reveals additional challenges in mapping scRNA-seq clusters to known cell types. To compare our data to the brain map, we used non-negative least squares regression to model each TAPIN-seq transcriptome as a linear weighted sum of single cell cluster transcriptomes, assuming that large regression coefficients reflect matching cell types (*Davie et al., 2018*; *Cao et al., 2019*). We interpreted the results of this comparison against an ideal scenario where single cell clusters were perfectly resolved and accurately labeled, and assuming that the driver lines used for TAPIN-seq profiling had minimal expression outside of the main target cell type. In this scenario, we would expect a unique cluster matching each of our TAPIN-seq profiles of cell type-specific driver lines, as well as many unmatched clusters, reflecting cell types that we did not profile. However, we observed few one-to-one matching clusters for our TAPIN-seq profiles (e.g., T1, Tm1, Dm8, Dm9, Pm3), several one-to-many matches (e.g., photoreceptor cluster #53 matching our R1-6, R7, R8-Rh5, and R8-Rh6 profiles; also, L1-5 cluster #20, Lawf1/2 cluster #58, and T4/T5 cluster #24), clusters with no TAPIN-seq matches (e.g., clusters #7, 15, 23), as well as TAPIN-seq cell types without a matching cluster (e.g., Dm4, Dm11, Mi4, Tm2, Tm20, LPLC1) (*Figure 5C*). The matches confirmed several clusters labeled as single cell types (e.g., Tm1, Mi1) or multiple cell types (e.g., photoreceptors, L1-5, Lawf1/2, T4/5, Kenyon cells) and also suggested possible labels for previously unlabeled clusters (e.g., Dm8 cluster #52, Dm9 cluster #74, pigment cell cluster #76) and alternative labels for previously labeled clusters (e.g., TmY5a TAPIN matches the TmY14 cluster #11; Lai TAPIN matches Dm8/Tm5c cluster #39). We observed similar results when analyzing the optic lobe map, with few apparent single cell – TAPIN-seq matches (*Figure 5—figure supplement 1*). Although this result could arise from major errors in our TAPIN-seq profiles, this possibility is unlikely given our earlier validation results and the concordance between our TAPIN-seq profiles and cell type-enriched genes identified from independent FACS-seq measurements (*Figure 2—figure supplement 2*).

As a separate comparison of the bulk and single cell profiles, we examined the bulk expression of genes marking each single cell cluster (*Figure 5—figure supplement 2*). Confirming the regression results, this analysis also found few one-to-one matches in which single cell cluster markers were enriched in only a single TAPIN-seq profile. Instead, most cluster markers were either enriched in multiple bulk cell types (over-clustering), or were not enriched in our data (cell types we did not profile by TAPIN-seq). As before, many TAPIN-seq profiles were not enriched for cluster markers, reflecting cell types that were either missing or clustered with other cell types in the single cell map.

We further explored specific examples where the TAPIN-seq data offered new insight into the single cell maps by suggesting alternative labels or labeling unannotated clusters. The single cell clusters labeled as TmY14 matched the TmY5a TAPIN profile (*Figure 5C*). The cluster was originally labeled based on the expression of a single transcription factor, *knot* (*kn*), as determined using a kn-

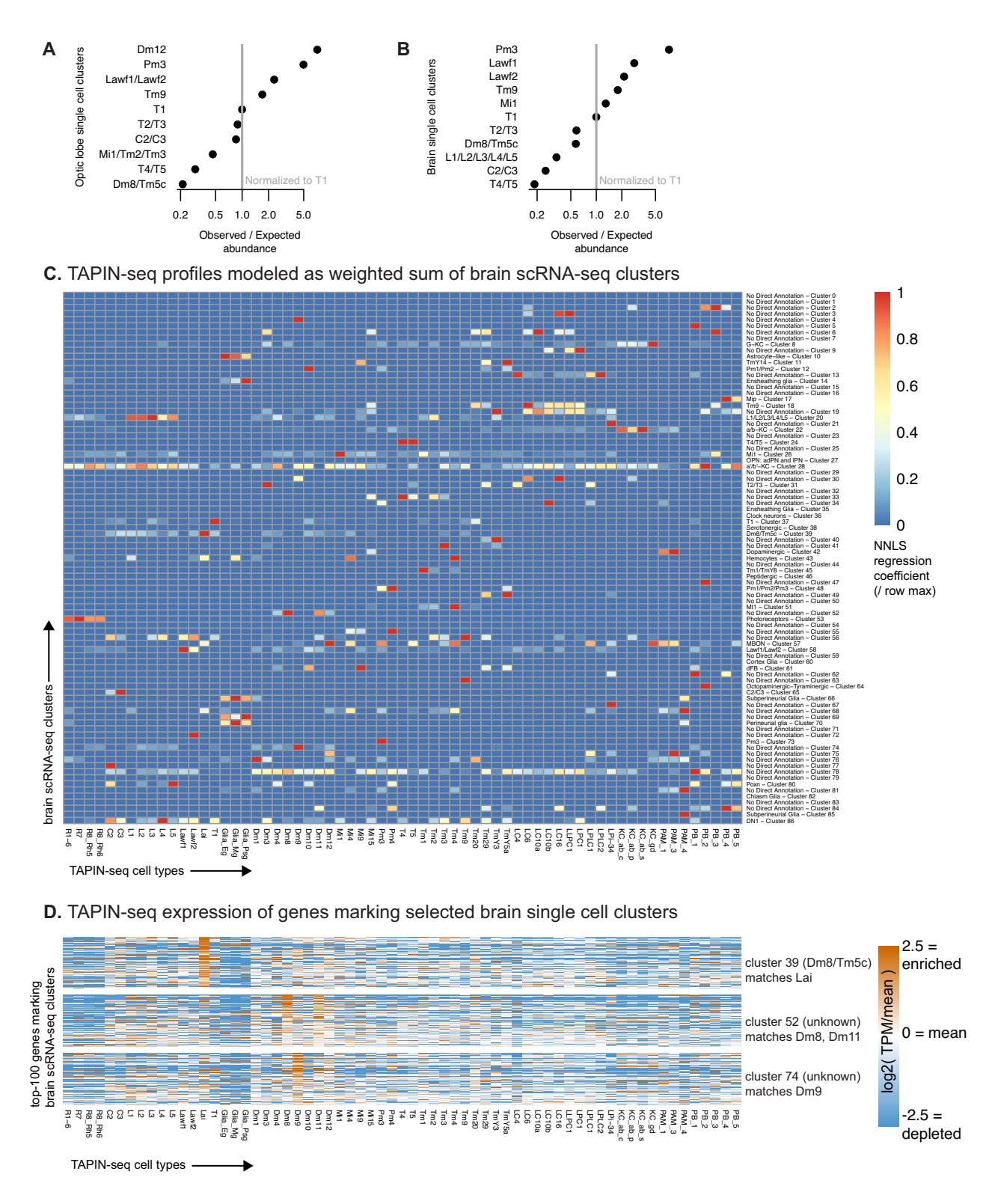

**Figure 5.** TAPIN-seq complements single cell RNA-seq profiling. (**A, B**) We evaluated whether single cell RNA-seq of the optic lobe (**A**) (*Konstantinides et al., 2018*) and brain (*Davie et al., 2018*) proportionally represents cell types found in the optic lobe. By comparing the single cell cluster sizes to the true abundance of each cell type (estimated as described in the Materials and methods) we found that the scRNA-seq map can both under- and over-estimate the abundance of each cell type (assuming accurate cell type labels), or that the cell type is incorrectly assigned (i.e. contains

*Figure 5 continued on next page*

*Figure 5 continued*

different or additional cell types). To estimate the true cell count, we made use of known anatomy (for example, several cell types are known to be present exactly once in each of the ~2×750 medulla columns per brain) or relied on published counts. In addition, we performed some new counts. (See Materials and methods for details.) Observed/expected ratio = ((size of cluster labeled as cell type X/size of cluster labeled as T1) / (true abundance of cell type X/true abundance of T1)). (C) We used non-negative least squares regression to model each TAPIN-seq profile as a linear weighted sum of single cell clusters in the whole brain scRNA-seq map. The heatmap represents the regression coefficients of each single cell cluster (rows) contributing to the TAPIN-seq profile of each cell type, normalized within rows. (D) We evaluated expression of genes that mark selected single cell clusters (*Davie et al., 2018*) in our TAPIN-seq profiles of visual system neurons. (see *Figure 5—figure supplement 2* for the complete heatmap). The online version of this article includes the following figure supplement(s) for figure 5:

**Figure supplement 1.** Regressing TAPIN-seq profiles against optic lobe single cell clusters.
**Figure supplement 2.** TAPIN-seq expression of genes marking single cell clusters.
**Figure supplement 3.** kn-GAL4 expression.

GAL4 reporter. We also observed *kn* expression in our TmY5a measurements and further confirmed its expression in both TmY14 and TmY5a cells using the kn-GAL4 reporter line, suggesting that the cluster likely includes not only TmY14 cells, but also TmY5a and other kn-expressing cells (*Figure 5— figure supplement 3*). Similarly, we found that the Dm2 cluster (optic lobe cluster #55), which was labeled based on a Dm2 FACS-seq profile, matched our Mi15 profile (*Figure 5—figure supplement 2A*). This observation is concordant with previous reports that the line used to FACS sort Dm2 also expresses in Mi15 (Supplementary Figure 2 in *Takemura et al., 2013*, Supplementary file 1 Table S4 in *Nern et al., 2015*). Finally, we found that the Dm8/Tm5c cluster (brain cluster #39) matches our Lai TAPIN-seq profile, while unlabeled clusters match our Dm8 and Dm9 TAPIN-seq profiles (*Figure 5D*). Our measurements also suggest labels for other previously unannotated clusters, such as brain cluster #76 which likely reflects pigment cell, as demonstrated by enrichment of its marker genes in both Dm1 and Tm20 profiles – both measured with lines that also express in pigment cells (*Figure 5—figure supplement 2B*). As expected, the genes marking this cluster include known pigment cell markers (e.g., *Pdh*, *rdhB*). Altogether, our results demonstrate that cell type-identified data, such as bulk transcriptomes, can help interpret single cell RNA-seq measurements.

## Profiles identify candidate neurotransmitter output for most neuron types

The proteins that synthesize and transport neurotransmitters are well known, enabling us to use their expression to predict neurotransmitter phenotype. We used histamine decarboxylase (*Hdc*), glutamate decarboxylase (*Gad1*), the vesicular acetylcholine transporter (*VAChT*), and the vesicular glutamate transporter (*VGlut*) to identify potential histaminergic, GABAergic, cholinergic, and glutamatergic cell types, respectively (*Figure 6A*). Our model unambiguously inferred expression states for these genes and indicated a single transmitter (from this group) for nearly all neurons we profiled. A second cholinergic marker, choline acetyltransferase (*ChaT*), matched *VAChT* expression almost perfectly (the two genes also share an exon). The sole exception, apparent expression of *ChAT* but not *VAChT* in R7 photoreceptors, likely results from a subset of dorsal rim R8 cells labeled by the R7 driver line (further discussed below, also see *Supplementary file 1A*).

Besides these four neurotransmitters that we identified by one or two marker genes, we also identified candidate dopaminergic neurons based on the combined expression of tyrosine hydroxylase (*ple*), dopa decarboxylase (*ddc*), vesicular monoamine transporter (*Vmat*) and dopamine transporter (*DAT*). While *DAT*, *ple,* and *ddc* were also expressed individually in several cell types that did not express *Vmat*, only known dopaminergic cell types and one medulla neuron (Mi15) expressed this combination (*Figure 6A*).

One neuronal cell type, T1, expressed none of the neurotransmitter markers *VGlut*, *VAChT*, *Vmat*, and *Gad1* (*Figure 6A*). Although T1 does express most pan-neuronal genes, it does not express bruchpilot (*brp*), a key component of presynaptic active zones. Consistent with this result, EM reconstruction has identified very few T1 presynaptic specializations (*Takemura et al., 2008*).

Transmitters for nearly half of our cell types have been previously proposed and generally agree with our results. For example, *VAChT/ChAT* expression in Kenyon cells supports recent reports showing they are cholinergic (*Barnstedt et al., 2016*; *Crocker et al., 2016*). Fluorescence in situ hybridization and immunolabeling guided by our measurements confirmed the expression of *ChAT*,

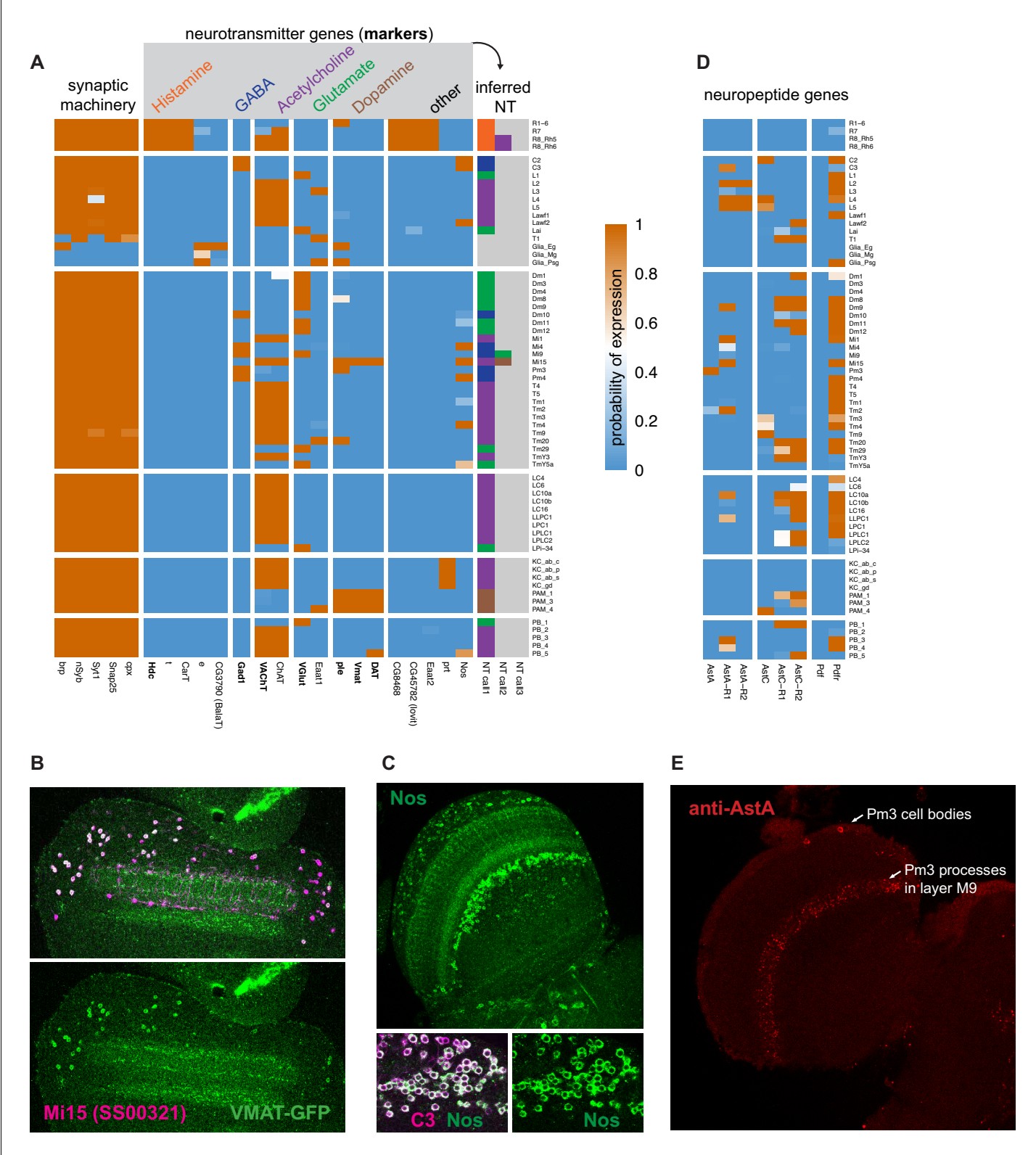

**Figure 6.** Expression of synthesis and transport genes indicate neurotransmitter phenotypes. (A) Expression of neurotransmitter marker genes indicate the neurotransmitters produced in nearly all profiled cells. With few exceptions, nearly all cell types express only one fast neurotransmitter. (B, C) We confirm TAPIN-seq results at the protein level (green) for (B) Vesicular monoamine transporter (Vmat) expressed in Mi15 (magenta) and (C) Nitric oxide synthase (Nos) in C3 (magenta). Top panel in (C) shows a section through the optic lobe, lower panels C3 cell bodies. (D) Several neuropeptides and

*Figure 6 continued on next page*

*Figure 6 continued*

receptors also express specifically (examples). (**E**) Allatostatin A (AstA) protein expression in the medulla as an example of a neuropeptide with a very specific optic lobe expression pattern. The AstA distribution in the optic lobe matches the distribution and layer pattern of Pm3 cells, consistent with the TAPIN-Seq data.

The online version of this article includes the following figure supplement(s) for figure 6:

**Figure supplement 1.** Transcriptional regulators of neurotransmitter identity.

*Gad1*, and *VGlut* in Mi1, Mi4, and Mi9, respectively (*Long et al., 2017*; *Takemura et al., 2017*). However, we see considerable differences between our assignments and some previous work that used reporter transgenes (*Raghu and Borst, 2011*; *Varija Raghu et al., 2011*; *Raghu et al., 2013*), which we generally attribute to unfaithful transgene expression patterns. We believe our assignments to be more reliable, however they are not without problems. For example, one assignment inferred by our model that seems unlikely and is not supported by other available data is the presence of *Gad1* in Mi9, which was not detected in the FISH or antibody experiments mentioned above. Given the presence of some contaminating Mi4 cells in at least one Mi9 driver and the lower *Gad1* abundance (mean 276 TPM in Mi9; 2165 TPM in Mi4; 1870 mean TPM in predicted GABAergic cells), we attribute the Mi9 *Gad1* signal to contaminating contributions from other GABAergic cells such as Mi4.

## Transcriptional regulation of neurotransmitter output

We next tried to identify transcriptional regulators of neurotransmitter output, by searching for TF genes expressed in strong correlation with transmitter phenotype. However, we only found such TFs for histaminergic output (*Figure 6—figure supplement 1A*), which in our dataset is only represented by photoreceptor neurons. This observation agrees with work on neuronal identity showing that single TFs rarely encode transmitter identity, but rather different TFs and TF combinations are used to specify the same neurotransmitter output (*Hobert, 2016*). We thus expanded our search to TFs whose expression was informative about transmitter phenotype (i.e., cells expressing TF A are likely to produce neurotransmitter B; even if not all cells producing neurotransmitter B express TF A; *Figure 6—figure supplement 1A*). This search identified candidate TFs for nearly all neurotransmitter types. For example, the 19 neuronal types (including the broad chat-GAL4 line) expressing apterous (*ap*) are cholinergic. Its worm ortholog, *ttx-3*, regulates the cholinergic phenotype of the AIY neuron (*Wenick and Hobert, 2004*). Several other TFs we identified also have worm or mouse orthologs implicated in neuronal identity (*Figure 6—figure supplement 1B*). Several TFs appeared to identify a transmitter phenotype within a group of cell types but not across the entire dataset. For example, Lim3 distinguishes the GABAergic Dm10 from the other Dm cell types in our dataset and is also expressed in several other GABAergic cells (Mi4, Pm3, Pm4) but was also detected in the cholinergic LC4 and the glutamatergic TmY5a and Tm29. We confirmed the differential *Lim3* protein expression in Dm10 and Dm12 cells (*Figure 6—figure supplement 1C*). Several of the transcription factors that we found to be informative of neurotransmitter output were also implicated by single cell RNA-seq data, including *ap* (cholinergic), *tj* (glutamatergic), and *Lim3* (GABAergic) (*Konstantinides et al., 2018*). Our data also indicate exceptions to these patterns (i.e., neurons expressing *tj* and *Lim3* but with a different neurotransmitter phenotype; *Figure 6—figure supplement 1A*). These observations indicate that neuronal features are likely regulated in a context-dependent and combinatorial manner, and that transcriptomes can identify putative regulators.

## Co-expression of canonical small molecule transmitters with non-canonical transmitters is widespread

Co-release of multiple neurotransmitters can enhance the signaling capacity of neurons and neural circuits. For example, the same cell type might release different transmitters under distinct conditions or use them to elicit distinct responses in different target cells. In addition to Mi9 (discussed above as being likely due to contamination), we observed two cases of potential co-transmission involving the canonical small molecule neurotransmitters. Both Mi15 drivers express dopaminergic and cholinergic markers, and both R8 drivers expressed cholinergic and histaminergic markers. We confirmed expression of Vmat protein in Mi15 (*Figure 6B*), the first identified columnar

dopaminergic cell type within the optic lobe, and further below we confirm the unexpected VAChT expression in R8 (Figure 8A).

Evidence for co-transmission involving additional molecules, such as neuropeptides or nitric oxide, appears frequently in our data set. Nitric oxide is a widely conserved signaling molecule that can act on many kinds of cells, including neurons (*Lowenstein and Snyder, 1992*). We observed very specific expression of its synthesizing enzyme, nitric oxide synthase (*Nos*), in the lamina (C2, C3, and Lawf2) and medulla (Mi4, Pm4, Tm4 and Mi15). To further validate these results, we confirmed Nos expression at the protein level in C3 neurons (*Figure 6C*). Nitric oxide can be released extra-synaptically, potentially enabling signaling between neurons that are not synaptic partners.

Several neuropeptides and their receptors were also expressed in distinct patterns suggesting widespread yet specific peptidergic signaling in the visual system (*Figure 6D*). For example, AstA was observed in just one cell type (Pm3; confirmed at the protein level; *Figure 6E*), while AstC was expressed in several cell types, and pigment-dispersing factor (Pdf) was expressed in none of the optic lobe cells we profiled. The receptors for all three of these neuropeptides were more broadly expressed (*Figure 6D*). The broad expression of *Pdf receptor (Pdfr)* is consistent with the extensive arborization previously observed for Pdf-expressing neurons at the surface of the medulla.

While we focused on genes with well-known functions, our expression patterns also suggest new functions for poorly characterized genes (*Figure 6A*). For example, photoreceptors specifically expressed *CG8468*, an orphan transporter in the solute carrier 16 (SLC16) family of monocarboxylate transporters. This gene might represent a candidate vesicular or plasma membrane transporter of histamine, which remains unidentified in any species. We also observed photoreceptor-specific expression of *CG45782* (*lovit*), a member of the SLC45 sucrose transporter family recently reported as a putative histamine transporter (*Xu and Wang, 2019*).

## Broad and patterned expression of neurotransmitter receptors

Since the functional consequences of the release of a neurotransmitter depend on which receptors for this transmitter are expressed in the receiving cell, measuring the expression of both neurotransmitter input and output genes is necessary to assign potential synaptic signs to connectomes. For example, glutamatergic transmission in *Drosophila* may be either inhibitory or excitatory, depending on the receptors.

In general, neurotransmitter receptors are broadly expressed, qualifying each cell type to detect multiple neurotransmitters (*Figure 7A*). Patterns for individual receptors (or receptor subunits) varied widely. Some receptors, such as the *GluClalpha* glutamate-gated chloride channel, thought to be the main mediator of inhibitory glutamatergic transmission in flies, were expressed in most but not all cell types (*Figure 7A,B*). Expression of others was much more restricted, such as the *EKAR* glutamate receptor subunit detected only in photoreceptor neurons, consistent with previous work (*Hu et al., 2015*). Nearly all cells expressed receptors for acetylcholine, GABA, and glutamate, as expected from the combination of predicted transmitter phenotypes and connectomics data. Receptors for neuromodulators such as serotonin, dopamine, octopamine, and neuropeptides in general were also widespread (*Figure 6D*). For example, octopamine receptors were expressed in broad, yet gene- and cell-type specific patterns, consistent with widespread octopaminergic modulation of visual processing (for example, *Arenz et al., 2017*; *Strother et al., 2018*; *Tuthill et al., 2014*). We confirmed Oamb expression at the protein level in specific lamina neurons and glia, including Lawf2 cells previously shown to be octopamine sensitive (*Tuthill et al., 2014*; *Figure 7C*).

## Combining transcriptomes and connectomes

A principal goal of our work is to provide a foundation for combining molecular data such as neuro-transmitter and receptor expression patterns with anatomical or functional connectivity data. One application of expression information is to constrain mechanistic models of neural circuits such as the extensively studied motion detection circuit in the fly eye (reviewed in *Mauss et al., 2017*; *Figure 7—figure supplement 1A*). The combined availability of expression and connectomics data for many cell types in a brain region also makes it possible to systematically identify and further explore unusual patterns of receptor or transmitter expression; for example, cell types in which an otherwise widely expressed receptor is absent or cells with unusual combinations of receptor subunits. Below we discuss three examples, focused on potential signs of synaptic transmission, of how such patterns

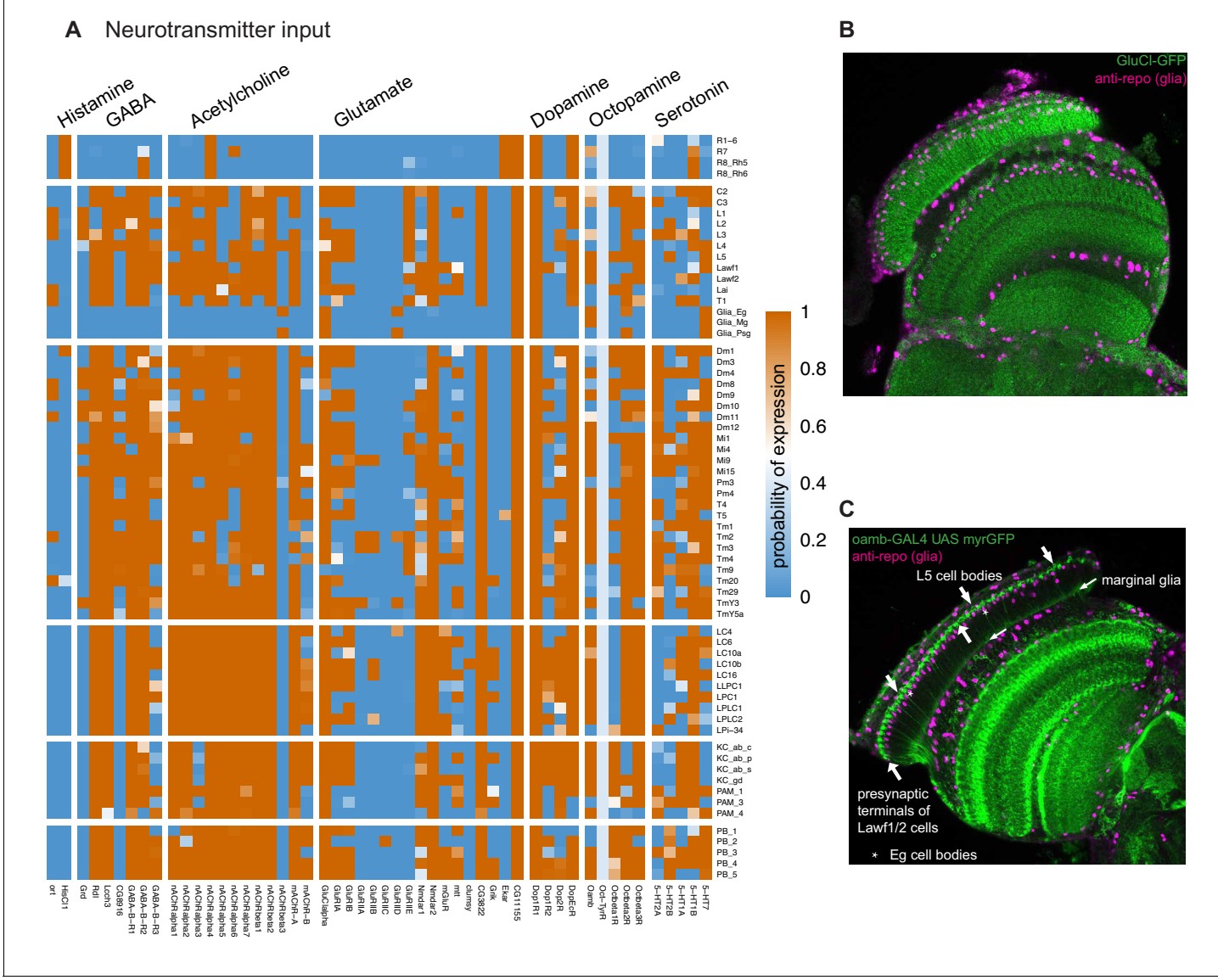

**Figure 7.** Patterns of neurotransmitter receptor expression. (**A**) Neurotransmitter receptors are widely expressed in specific patterns. With the exception of histamine, most cells express receptors or receptor subunits for nearly all neurotransmitters. (**B**) Expression of the glutamate-gated chloride channel (GluClalpha), detected using a GFP-tag (green), in the optic lobe. The lamina pattern includes many neurons as well as proximal satellite, epithelial and marginal glia. A glia-specific nuclear marker (anti-repo) is shown in magenta. (**C**) Octopamine receptor (Oamb) expressing cells in the optic lobe detected with a protein-trap GAL4 driving expression of a membrane targeted GFP (green). Anti-repo (magenta). In the lamina (to the top and left of the image), Lawf1/2 and L5 neurons and marginal glia are recognizable.

The online version of this article includes the following figure supplement(s) for figure 7:

**Figure supplement 1.** Patterns of neurotransmitter receptor expression complement connectomics.

can lead to specific and unexpected hypotheses about circuit function. As we illustrate, combining expression data with synapse-level anatomy permits analyses which are inaccessible to either approach alone.

### Presynaptic cholinergic markers and absence of histamine receptors in some postsynaptic targets: R8 photoreceptors may signal via two fast transmitters

Fly photoreceptors have long been known to release histamine (*Hardie, 1987*; *Sarthy, 1991*). Our data indicate that inner (color vision) R8 photoreceptors also express the cholinergic markers *ChAT* and *VAChT*, suggesting an unexpected additional cholinergic phenotype (*Figure 6A*). To confirm these results, we visualized a tagged VAChT protein (VAChT-HA; *Pankova and Borst, 2017*), expressed from the endogenous locus, selectively in photoreceptor cells. These experiments showed

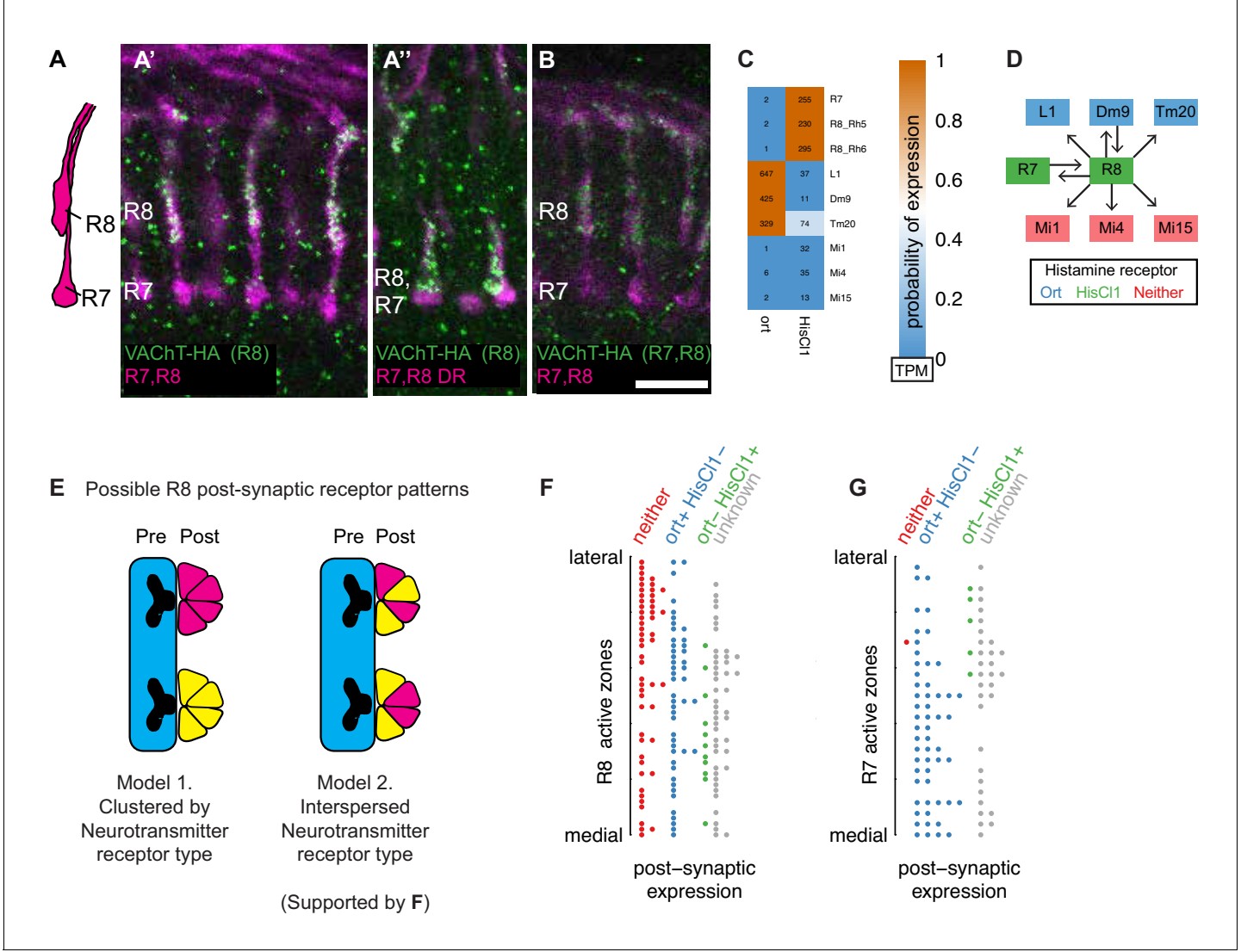

**Figure 8.** Molecular and connectomics analyses suggest R8 photoreceptors signal via both histaminergic and cholinergic neurotransmission. (A, A', A'', B). Expression of VAChT in R8 cells. Expression of a HA-tagged VAChT was induced in R8 cells by recombinase-mediated excision of an interruption cassette from a modified genomic copy of the VAChT gene (*Pankova and Borst, 2017*). R7 and R8 cells project to different layers of the medulla (A, schematic). Single confocal sections show R7 and R8 cells in magenta and anti-HA immunolabeling in green. R7 and R8 cells (labeled with mAb 24B10) are shown in magenta. Stop-cassette excision in R8 photoreceptors (using sens-FLP) results in VAChT-HA labeling of R8 terminals in both the main medulla (A') and the dorsal rim (where R7 and R8 cells project to very similar layer positions) (A''). Stop-cassette excision in all photoreceptors (using ey3.5-FLP) also produces VAChT-HA labeling in R8 while expression in R7 was not detected (B). Scale bar, 10 µm. (C) Heatmap of receptor expression probabilities (color) and relative abundance (numbers; transcripts per million) in R8 targets identified by EM (at least five synapses in *Takemura et al., 2013*). (D) Connectivity network for R8 cells, overlaid with receptor expression. (E) Possible distributions of postsynaptic receptors at R8 synapses. Individual active zones can interact with multiple postsynaptic cells which could be grouped in distinct ways. (F) Classification of postsynaptic cells at individual R8 active zones (*Takemura et al., 2013*) based on histamine receptor expression. (G) Same analysis as in F but for an R7 cell.

VAChT-HA labeling in medulla terminals of R8 but not R7 cells (*Figure 8A,A',A'',B*), including the specialized polarized light-responsive R8-cells in the dorsal rim of the medulla. The latter express the rhodopsin Rh3 (which is otherwise expressed in R7s; *Fortini and Rubin, 1990*), consistent with the presence of ChAT and VAChT transcripts in the R7 driver line (for which the model inferred expression for VAChT but not ChAT).

We asked whether the apparent co-transmitter phenotype of R8 neurons was reflected in the expression of neurotransmitter receptors in their different postsynaptic partners. Postsynaptic partners of R8 cells identified by electron microscopy reconstructions include seven cell types in our dataset: Dm9, Mi1, Mi4, Mi15, R7, L1 and Tm20 (*Figure 8C*; *Takemura et al., 2013*; *Takemura et al., 2015*). All of these express one or more nAChR subunits (*Figure 7A*). By contrast, expression of the histamine-gated chloride channels *HisCl1* and *ort*, which mediate histaminergic transmission by photoreceptors (*Pantazis et al., 2008*), was more selective (*Figure 8C,D*): L1, Tm20 and Dm9 express *ort*, consistent with previous reports (*Gao et al., 2008*), while *HisCl1* transcripts were detected in the R7 as well as R8 driver lines, in agreement with another recent report (*Schnaitmann et al., 2018*; *Tan et al., 2015*). However, we did not find evidence of expression of *ort* or *HisCl1* in Mi4, Mi1 and Mi15, further supporting R8 signaling via a transmitter other than histamine.

We were interested in whether release of ACh and histamine might occur at spatially distinct locations. Insect synapses often consist of multiple postsynaptic sites apposed to the same presynapse. For cells that release more than one transmitter, two general distributions of postsynaptic processes at such multicomponent synapses are possible (*Figure 8E*). Postsynaptic cells with different receptors could be grouped at different sites based on receptor expression (*Figure 8E*-left) or occur together at the same locations (*Figure 8E*-right). To distinguish these possibilities for R8 cells, we used EM reconstruction data (*Takemura et al., 2013*) to map the predicted expression of histamine receptors in postsynaptic cells at the single synapse level for all presynaptic sites of one reconstructed R8 cell (*Figure 8F*). The resulting pattern indicates that processes of cell types with and without histamine receptor expression are often located near the same R8 presynapse (*Figure 8F*), whereas this is not the case for a reconstructed R7 cell (*Figure 8G*). This is consistent with the VAChT-HA labeling observed throughout the medulla terminals of R8s (*Figure 8A*). This spatial pattern is compatible with either co-release of histamine and ACh or independently regulated release from different vesicles at the same sites.

A combined cholinergic and histaminergic phenotype has been reported for a small group of extraretinal photoreceptors (the Hofbauer-Buchner eyelet) located near the lamina (*Yasuyama and Meinertzhagen, 1999*) but was unexpected for R-cells of the compound eye. Establishing the functional significance of potential acetylcholine release by R8 cells will require further experiments. However, we note that double mutants lacking both histamine receptors are not completely blind (*Gao et al., 2008*), consistent with histamine-independent transmission by photoreceptor neurons. In addition, a very recent study suggests a role of cholinergic R8 signaling in the entrainment of the fly's circadian rhythm to light-dark cycles (*Alejevski et al., 2019*), perhaps similar to that of ACh-release from the Hofbauer-Buchner eyelet (*Schlichting et al., 2016*).

## Potentially excitatory GABA-A receptors in lamina monopolar cells

Fast GABAergic transmission via GABA-A receptors is a major source of inhibition in the nervous system. However, some GABA-A subunit combinations could mediate depolarizing GABA-signaling: in vitro assays indicate that homomeric Rdl or heteromeric Rdl/Lcch3 receptors are typical GABA-gated chloride channels (*Zhang et al., 1995*), while Lcch3/Grd form GABA-gated cation channels (*Gisselmann et al., 2004*). However, the in vivo significance of this difference is unknown. *Rdl* and *Lcch3* were expressed in nearly all neurons in our dataset (*Figure 7A*, *Figure 9A,B*), consistent with the general inhibitory nature of GABA signaling. By contrast, *Grd* and another predicted GABA-A receptor subunit, *CG8916*, were expressed in a minority of cell types (*Figure 7A*, *Figure 9A,B*). Photoreceptor neurons, for which no major GABAergic inputs have been identified by connectomics, expressed none of the four transcripts (*Figure 9B*). Lamina monopolar L1 and L2 were the only neurons other than photoreceptors that did not express significant levels of *Rdl*. However, both express *Grd*, *Lcch3* and also *CG8916*. Together with the in vitro findings mentioned above, this result suggests that some or all GABA-A receptors in L1 and L2 may be cation rather than chloride channels.

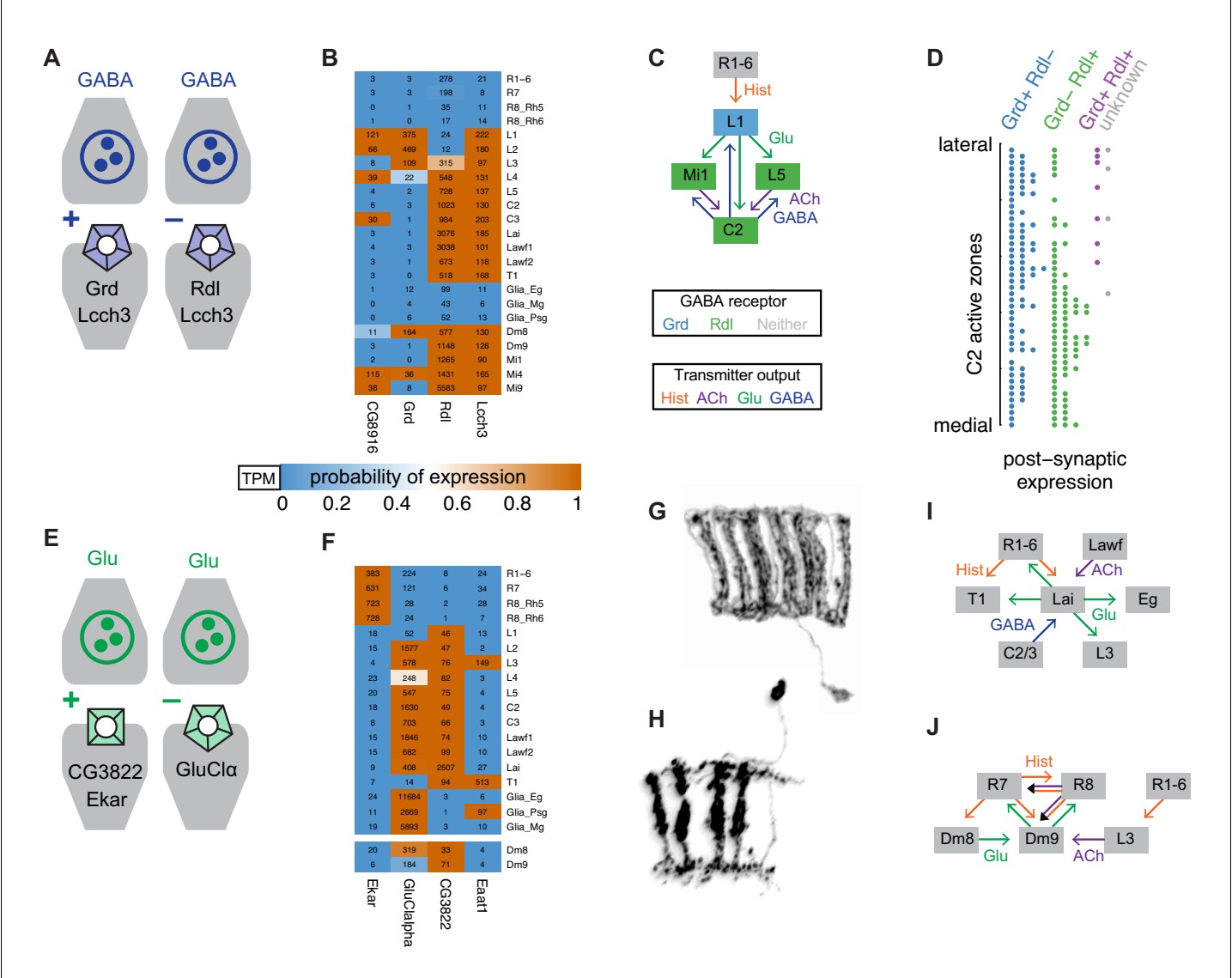

**Figure 9.** Using gene expression to functionally interpret circuit structure. (A) Different properties of GABA-A receptors in *Drosophila* observed in in vitro studies. GABA-A receptor subunits can form either cation or anion channels depending on subunit composition. (B) Expression of GABA-A subunits in selected cell types. (C) L1 and two of its target cells form strong reciprocal connections with C2 neurons. (D) Distribution of *Rdl* and *Grd* expressing cells at individual C2 synapses. (E) Glutamate receptors can also be excitatory or inhibitory. (F) Examples of expression patterns for selected glutamate receptors and transporters. (G, H) Morphology of Lai (G) and Dm9 (H) cells. Illustrations based on MCFO images of single cells. (I, J) Analysis of the input and output pathways of Lai (I) and Dm9 (J) neurons suggests a potentially similar functional role for these cells. The predicted absence of GluCl-alpha in Dm9 suggests that glutamatergic input from Dm8 to Dm9 may be excitatory.

Remarkably, lamina monopolar cells in the housefly *Musca,* which are thought to have very similar functional properties to those in *Drosophila*, depolarize in response to GABA (*Hardie, 1987*) but hyperpolarize in response to histamine (via ort-containing chloride channels). Thus our data identify a potential link between in vivo electrophysiology, in vitro receptor properties and cell type differences in GABA-A subunit (Rdl or Grd) expression.

We next asked whether depolarizing GABA-signaling to L1 and L2 was plausible given their synaptic connectivity. Based on synapse counts and our transmitter data, the main GABAergic inputs to L1 and L2 are C2 and C3 neurons (*Meinertzhagen and O'Neil, 1991*; *Rivera-Alba et al., 2011*; *Takemura et al., 2013*; *Takemura et al., 2015*). Conversely, L1 is the main input to both C2 and C3 cells, followed by the cholinergic L1 targets L5 and Mi1. These strong connections (illustrated for C2

in *Figure 9C*) indicate that the effective sign of GABA input to L1 and L2 is almost certainly of functional significance. In the illustrated circuit (*Figure 9C*), L1 cells hyperpolarize in response to luminance increases (as histamine from photoreceptors opens ort chloride channels). The resulting reduced secretion of glutamate is thought to depolarize L1 targets such as Mi1 (via closing of *GluClalpha* channels). One plausible, though speculative scenario, is that similar to Mi1, C2 cells also depolarize in response to light. In this case, GABA-gated cation channels in L1 (formed by Grd and Lcch3) would enable negative feedback (counter-acting) from C2 to L1, which for example could return the membrane potential closer to resting levels – speeding up the response to subsequent luminance changes. By contrast, opening of conventional GABA-A receptors (GABA-gated chloride channels) in L1 would resemble a light response (opening of histamine-gated chloride channels), and thus provide positive (reinforcing) feedback in this case. The latter possibility appears less consistent with the transient nature of the L1 (and L2) response to light (*Jarvilehto and Zettler, 1971*; *Laughlin and Hardie, 1978*). Distinguishing these and other possibilities will of course require future experimental work.

Similar to the findings for histamine receptors described above (*Figure 8F*), again using connectivity data for the medulla, we observed that cells with different GABA-A profiles can be postsynaptic at the same synapse (*Figure 9D*). In addition to L1 and L2, *Grd* expression indicated several other candidates for cells with unusual GABA responses (*Figures 7A* and *9B*). In these neurons (e.g., Dm8 or Mi4), *Rdl* and *Grd* were detected together, raising questions such as whether their subcellular distribution is synapse-specific or whether these subunits might co-assemble into channels with yet unexplored properties.

## Diverse patterns of glutamate receptor expression in the targets of a single local interneuron type in the lamina

The diverse expression of glutamate receptor subunits, which can mediate both inhibitory and excitatory signaling, was particularly striking in the lamina (*Figures 7A* and *9E,F*). Notable patterns include the photoreceptor-specific expression of *EKAR*, the predicted absence of *GluClalpha*, otherwise broadly expressed in neurons, from some cell types, including photoreceptors (*Figures 7A* and *9F*) and its strong expression in epithelial glia. *CG3822*, a Kainate-type receptor subunit recently reported to function in presynaptic homeostatic control at the neuromuscular junction (*Kiragasi et al., 2017*), was strongly enriched in the lamina intrinsic Lai cells. Since Lai neurons are the only known source of vesicular glutamate release in the lamina, *CG3822* function in Lai is predicted to also be pre- or perhaps extrasynaptic. T1 and L3, while not expressing *VGlut*, might also influence glutamate levels in the lamina via the *Eaat1* plasma membrane glutamate transporter. The strong expression of this transporter in T1 rather than glia is another unusual feature of glutamatergic signaling in the lamina and may be a clue to the enigmatic function of T1 cells (*Tuthill et al., 2013*). These examples further highlight how a transmitter released by one neuron type, here Lai, is predicted to have very different effects on target cells due to the receptors they express.

## Comparisons of cell shape, synaptic connectivity and receptor expression reveal multiple similarities between local interneurons Lai in the lamina and Dm9 in the medulla

The combination of highly specific *EKAR* expression and the unusual absence of *GluClalpha* in photoreceptors prompted us to further explore cellular sources and potential functions for glutamatergic signaling to photoreceptor neurons.

Photoreceptor neurons function over an extremely wide range of light levels, from moonlight to bright sunlight. One mechanism proposed to enable this behavior is a depolarizing feedback signal from photoreceptor targets that increases photoreceptor output under low light conditions, but reduces output at higher light intensities (*Zheng et al., 2009*). As Lai cells express *ort*, and thus, like other *ort*-expressing photoreceptor targets, are thought to hyperpolarize in response to light, increased glutamate release from Lai could provide such light-dependent feedback via EKAR in R-cells. This scenario is consistent with reduced photoreceptor responses at low light intensities after reduction of Lai output or EKAR function (*Hu et al., 2015*). *EKAR* is also expressed in R7 and R8 cells, which project to the medulla and are not postsynaptic to Lai. We therefore asked whether there might be a medulla counterpart of Lai neurons.

Synaptic connectivity data identify Dm9 as a strong candidate for such a role: Dm9 is both a major pre- and postsynaptic partner of R7 and R8; it is the only identified R7/R8 target with these properties (other known R7 or R8 targets appear to form few if any feedback synapses on these cells). Remarkably, the overall anatomy of Dm9 cells is also very similar to Lai (*Figure 9G,H*): Both Lai and Dm9 cells span multiple visual columns but the precise number and distribution of columns innervated by each individual cell is variable. Finally, Lai and Dm9 share key molecular properties: for example, both are glutamatergic and express *ort* histamine receptors. Based on connectivity and gene expression (*Figure 9F,J*), Dm9 cells are predicted to receive hyperpolarizing R7 and R8 input via ort and excitatory input from the photoreceptor targets L3 and Dm8. Thus, similar to Lai (*Figure 9G,I*), Dm9 appears qualified to increase photoreceptor output in the medulla under low light conditions, similar to a proposed function of Lai in the lamina.

One notable difference between Lai and Dm9 is that in contrast to Lai, Dm9 cells receive input from photoreceptor neurons with different spectral tuning. This input involves direct (R7, R8) and indirect pathways (R7 via Dm8, R1-6 via L3) (*Figure 9J*). This integration of multiple spectral inputs could support a role of Dm9 in color processing. Indeed, the anatomical and predicted functional properties of Dm9 match those of an as yet unidentified *ort* expressing cell type proposed to contribute to color opponent signaling between R7 and R8 cells (*Schnaitmann et al., 2018*).

## Discussion

We present an approach to characterize the function of neural circuits by combining genetic tools to access their component cells, TAPIN-seq to measure their transcriptomes, and a probabilistic model to interpret these measurements (*Figures 1*, *2* and *3*). We used this approach to establish an extensive resource of the genes expressed in 67 *Drosophila* cell types, including 53 in the visual system, covering photoreceptors, lamina, and components of the motion detection circuit (*Figure 4*) and systematically compare our results to single cell RNA-seq (*Figure 5*). Our approach enables an extensive analysis of neurotransmission in the *Drosophila* visual system, including the neurotransmitters sent and received across the network as well as transcription factors that potentially regulate neurotransmitter identity (*Figures 6* and *7*). We also provide specific examples of integrating transcriptomes and connectomes to illuminate circuit function (*Figures 8* and *9*).

Many recent studies have explored gene expression in neurons. However, only a few of these were aimed at neurons in genetically tractable organisms and brain regions for which detailed anatomical data, especially at the level of synaptic connections, are available. Previous work in the mouse retina has used both genetic (*Siegert et al., 2012*) and single cell approaches (*Macosko et al., 2015*) to characterize transcriptional regulators as well as classify cell types. More recent work in *Drosophila* used single cell RNA-sequencing to characterize heterogeneity in olfactory projection neurons (*Li et al., 2017*), the midbrain (*Croset et al., 2018*), the optic lobe (*Konstantinides et al., 2018*), and the whole brain (*Davie et al., 2018*). The expression patterns of many genes have also been mapped in *C. elegans* neurons, whose connectivity has long been known, although these studies typically focus on individual genes rather than genome-wide catalogs (*Hobert, 2016*). The unique combination of an extensive genetic toolbox to access individual cell types in the *Drosophila* visual system and systematic efforts to map its connectivity, make it well suited for exploring whether a comprehensive catalog of gene expression is useful for understanding circuit function. Towards this end, we profiled a diverse array of cell types including all of the neuronal cell types that populate the lamina and a subset of cell types in the medulla and lobula complex including those known to play a central role in the detection of motion. We also analyzed a number of cell types residing in deeper brain structures such as the mushroom body and central complex.

Our approach requires genetic driver lines to obtain transcriptomes of specific cell populations. The recent availability of large collections of reagents for split-GAL4 intersections (*Dionne et al., 2018*; *Tirian and Dickson, 2017*) make it possible to obtain such lines for virtually any cell type of interest. This expanding genetic toolbox works well with our TAPIN-seq method to profile transcriptomes.

In some cases, available driver lines, including some used in this study (see *Supplementary file 1A*), may label some additional cell types. In addition to the presence of different, anatomically distinct cell types in a driver pattern, heterogeneity could also result from as yet unrecognized subpopulations within a seemingly uniform group of cells. For example, R7 and R8 photoreceptor neurons

each include two major subtypes (pale and yellow) with different rhodopsin expression but very similar, if not identical, cell morphology (*Wernet and Desplan, 2004*). While drivers with even higher specificity could be obtained through testing of additional split-GAL4 intersections or perhaps triple intersections (*Dolan et al., 2017*), we did not find the contributions of small numbers of 'off-target' cells to be a major limitation for many applications of expression data. In general, the transcriptomes support the high specificity of the intersectional lines we used to access visual system cells (*Figure 1*). For example, we found specific expression of known marker genes (*Figures 2G* and *4B*) and also that most neurons only express genes for a single neurotransmitter type (*Figure 6A*). In general, the expression patterns observed in our validation experiments (for example, *Figure 3G* and *Figure 3— figure supplement 2*) were highly consistent within a cell type. This suggests that individual cells of many if not all of the anatomical cell types we profile do indeed share specific molecular signatures, even if subdivisions within some types might exist. The availability of specific driver lines makes such validation at cellular resolution possible in a way that is otherwise difficult, for example in single cell RNA-seq studies. Driver lines also permit repeated access to the same cell type in multiple animals at defined time points, enabling the study of behavioral or circadian conditions in individual cell types without having to sequence the whole brain or dissected brain regions.

Modifying the one-step affinity capture in the original INTACT method to a two-step capture in TAPIN-seq increased its specificity, sensitivity, and throughput without the need for time-consuming and labor-intensive centrifugation steps (*Figure 1*). We initially tried improving the original INTACT method by using density gradient centrifugation to purify nuclei prior to the bead capture step, but this was cumbersome, low throughput, and ineffective for cell types with few nuclei per brain. In addition, for reasons that remain unclear, both photoreceptors and T4 cells consistently yielded few nuclei with this approach. Even with TAPIN, the libraries obtained with some sparser driver lines did not meet the quality control standards we applied. We suspect that the quality of these sub-optimal libraries can be improved by starting with more flies, which is simplified by TAPIN-seq's ability to use frozen material, enabling the collection of many flies on multiple days at defined time points. In contrast, manual or FACS sorting of dissociated cells is more challenging to scale up, because these more labor-intensive tissue procurement schemes cannot be simplified in the same way. It is also worth noting that our tandem affinity purification approach can improve the specificity of any immunopurification method that uses a capture antibody that is cleavable by IdeZ (all IgG subclasses), without requiring expression of a traditional TAP tag (*Rigaut et al., 1999*).

TAPIN-seq complements single-cell RNA-seq studies of neurons in several ways (*Ecker et al., 2017*; *Konstantinides et al., 2018*). First, our high-resolution transcriptomes will serve as a reference for interpreting single-cell measurements. In particular, comparing our expression catalog to recent single cell maps of the optic lobe and whole brain highlights the challenges in interpreting single cell measurements. Several cell types that we profiled don't appear as clusters in the single cell map, while others are grouped into the same cluster. The well-established neuroanatomy of the optic lobe makes it an ideal setting to evaluate the accuracy of single cell RNA-seq measurements and raises a broader caution when interpreting scRNA-seq surveys of less well-characterized tissues: the composition of cell types (or states or clusters) observed by scRNA-seq can deviate significantly from their true abundance and requires validation with independent methods. While we analyze fly neurons here, the cautions may also be worth considering in other tissues and species (e.g., the Human Cell Atlas effort; *Regev et al., 2017*). Having both deep bulk transcriptomes and single cell maps of the same tissue also provides an opportunity for developing new analytical tools that can harness available cell type-identified information while clustering single cell data. Second, combining our approach with single-cell profiling could more efficiently profile heterogeneity within a brain region or genetically defined cell population. Such a combined approach could also help to characterize known cell types for which specific driver lines are not yet available or help to identify and exclude contributions from 'off-target' cells to a bulk profile. Finally, the complementarity between bulk and single-cell measurements extends to other genomic features that can be measured in TAPIN-seq purified nuclei, including accessible chromatin and modified histones. We expect this combination of genomic tools to help decipher the transcriptional and epigenetic regulation of neuronal expression programs.

Transcriptome measurements can be of limited utility because it is challenging to interpret relative transcript abundance. In this study we developed a probabilistic mixture modeling approach to classify relative abundances into binary on and off states. This model was a useful guide for

interpreting our measurements; most genes are readily described with the two-state model, although the expression of some genes is not (e.g., *Rab11*; *Figure 3—figure supplement 1L*). Even for specific genes whose expression is more continuous than bimodal, the results still offer a useful family-wide summary of expression patterns. For example, DPR family members are more broadly expressed than DIP genes (*Figure 4C*; *Figure 4—figure supplement 1D*), an observation supported by two recent studies using different methods (*Cosmanescu et al., 2018*; *Venkatasubramanian et al., 2019*). Despite our model's utility, it is important to remember the many potential sources of error (minor cell types in driver line patterns, transcript carry over during TAPIN, biases in RNA-seq library construction and sequencing, etc.) that can affect measurements of relative transcript abundance and the resulting model inferences. Having observed most discrepancies between our modeling results and protein-level expression near the boundary between on and off states, it is prudent to treat these cases more carefully.

Our resource provides additional foundation for systematic functional and molecular studies of the *Drosophila* visual system. We illustrated how the resource can characterize neurotransmission in the network, particularly when combined with connectome information detailing connectivity between cell types as well as the grouping of post-synaptic partner cell types. We predict neurotransmitters used by every cell we profiled and found two likely cases of co-transmission (*Figure 6A*). The expression patterns of the major fast-acting transmitters histamine, acetylcholine, glutamate and GABA were comparatively simple: Nearly all neuronal cell types in our catalogue appear to express exactly one of these four transmitters. However, the transcriptomes suggest that many cells also have the potential to release specific neuropeptides, other chemical messengers such as nitric oxide, or form gap junctions with other cells.

While selected transmitter markers (e.g., Gad1 or VGlut) could also be assigned to cell types using methods such as immunolabeling or FISH, these approaches are not practical for comprehensive sampling of markers across these different modes of cell-cell communication. This is particularly clear when the expression patterns of neurotransmitter receptors are also considered (*Figure 7A*). Our results suggest that, for canonical small molecule transmitters, neurotransmitter output space is tightly tuned while input space is not: neurons typically send just one type of signal but can receive many (*Figures 6* and *7*). The expression patterns of neurotransmitter receptors provide further context for determining circuit mechanisms (*Figure 7—figure supplement 1*, *Figures 8* and *9*). Our results also implicate transcription factors involved in regulating neurotransmitter phenotype, including several that appear to have conserved roles in specifying neuronal identity in other species (*Figure 6—figure supplement 1*).

The availability of connectivity data for many neurons in the visual system allowed us to interpret neurotransmitter use and receptor distribution in the context of circuit architecture (*Takemura et al., 2013*; *Takemura et al., 2015*; *Rivera-Alba et al., 2011*). For example, our data show expression of both histaminergic and cholinergic markers in R8 photoreceptors. We further find that some major synaptic targets of R8 cells, as identified by electron microscopy, do not express known receptors for histamine (*Figure 8F*). Given that R7 and R8 cells were previously thought to be exclusively histaminergic both results are unexpected and individually might appear difficult to explain (*Gao et al., 2008*). However, in combination, these findings make a strong case for a dual histaminergic and cholinergic transmitter phenotype of R8 cells. In contrast, R7 only expresses the histaminergic marker *Hdc*, and all of its targets express a histamine receptor (*Figure 8G*).

Finally, our approach especially complements ongoing efforts to map circuit connectivity, which is complete for *C. elegans*, and is becoming accessible on a whole brain level for *Drosophila* (*Zheng et al., 2018*), and for portions of the mouse brain such as the retina. Methods to obtain and interpret serial electron micrographs, array tomography and other methods for mapping connectivity are rapidly progressing (*Swanson and Lichtman, 2016*; *Micheva and Smith, 2007*; *Kebschull et al., 2016*). All told, we are entering a period in neuroscience where connectomics will become pivotal. We expect that genomic approaches, such as the methods for data collection and analysis that we describe here, will enhance these efforts by using transcriptomes to provide, at high-throughput, a molecular proxy for physiological features that are otherwise inaccessible to connectomic methods.

## Materials and methods

All reagents and other resources are listed in the Key Resource Table (*Supplementary file 2*). A detailed description of split-GAL4 hemidrivers (https://bdsc.indiana.edu/stocks/gal4/split_intro.html) and cell-type specific split-GAl4 lines is available online (https://www.janelia.org/split-GAL4).

### Experimental models and subject details

Flies were reared on standard cornmeal/molasses food at 25°C. For profiling experiments adults, 4–7 days of age, were entrained to a 12:12 light:dark cycle and anesthetized by $CO_2$ at ZT8 - ZT12. Samples can be stored indefinitely at $-80°C$ after flash freezing in liquid $N_2$. We used female flies for all anatomical characterizations.

### Method details

#### Anatomical analyses

Details of individual genotypes and labeling methods used in the characterization of the driver lines and other anatomical experiments are summarized in *Supplementary file 1E*. Details of the driver lines are provided in *Supplementary file 1A*. For the naming of RNA-seq samples, we identified all drivers with a main cell type or cell types (e.g., Mi9_d1). Most of these cell types have been described in detail and were identified based on prior descriptions (for details see *Supplementary file 1A*; *Takemura et al., 2013*; *Gao et al., 2008*; *Nern et al., 2015*; *Fischbach and Dittrich, 1989*; *Tuthill et al., 2013*; *Aso et al., 2014*; *Wu et al., 2016*; *Wolff and Rubin, 2018*; *Wolff and Rubin, 2018*; *Edwards et al., 2012*; *Helfrich-Förster et al., 2007*; *Panser et al., 2016*; *Mauss et al., 2015*). The driver names do not attempt to include additional cells present in some drivers. A few of our cell types are strictly groups of related cell types (for example, the muscle cells or, at a different level of a cell type hierarchy, the T4 and T5 cells, with four subtypes each, or R7 photoreceptor neurons, which include R7s of pale and yellow ommatidia).

#### Generation and characterization of new driver lines

Split-GAL4 and GAL4 driver lines (*Supplementary file 1A*) were used to express UNC84-2XGFP in defined cell populations. Previously published driver lines were from the following studies (see *Supplementary file 1A* for details; *Tuthill et al., 2013*; *Diao et al., 2015*; *Aso et al., 2014*; *Wu et al., 2016*; *Strother et al., 2017*; *Park et al., 2003*; *Rulifson et al., 2002*; *Tayler et al., 2012*; *Taghert et al., 2001*; *Brand and Perrimon, 1993*; *Wu et al., 2003*; *Park et al., 2000*; *Sweeney et al., 1995*; *Wolff and Rubin, 2018*; *von Reyn et al., 2017*). New split-GAl4 lines were generated as in previous work (*Tuthill et al., 2013*; *Wu et al., 2016*). Briefly, we first identified GAL4 lines with expression in the cell type of interest by screening images of the expression patterns of large collections of such lines (*Jenett et al., 2012*; *Tirian and Dickson, 2017*). Typically, several candidate combinations of AD- and DBD-hemidrivers were tested to identify lines with sufficient specificity.

To characterize new driver lines, we examined both overall expression pattern in the brain and optic lobe and, for most lines, confirmed the identity of the main cell type or types using MultiColor FlpOut (MCFO)-labeled single cells (*Nern et al., 2015*). Since details of the expression patterns of GAL4 or split-GAL4 driver lines can depend on the particular UAS reporter used, we re-imaged 20 drivers with the TAPIN nuclear marker used for the profiling experiments (*Figure 2A*). In general, the distribution of labeled nuclei in these images appeared to match the expression patterns and specificity expected from the driver line's original characterization using a membrane marker. As expected, a small number of off-target cells were detectable (often more weakly labeled) in many driver lines.

#### Validation experiments

For validation experiments, we examined expression patterns of tagged proteins expressed in a near native genomic context using either large BAC-transgenes or modifications of the endogenous loci (*Nagarkar-Jaiswal et al., 2015*; *Diao et al., 2015*; *Kudron et al., 2018*; *Lee et al., 2018*).

We classified fkh-GFP and Ets65A-GFP as expressed or not expressed by visually comparing nuclear GFP signal in cells of interest (identified using a split-GAL4 driver) to background labeling in surrounding cells. Because of considerable differences in the GFP signal for different cell types,

confocal settings and post-imaging adjustments were done individually for different cell types for these experiments.

The following transgenes were used (also see *Supplementary file 1E*):

PBac{y[+mDint2] w[+mC]=fkh GFP.FPTB}VK00037 (RRID:BDSC_43951), PBac{y[+mDint2] w[+mC] =Ets65 A-GFP.FLAG}VK00037 (RRID:BDSC_38640), Mi{PT-GFSTF.0}Nos[MI09718-GFSTF.0] (RRID: BDSC_60278), Mi{Trojan-GAL4.1}Oamb[MI12417-TG4.1] (RRID:BDSC_67506), Mi{Trojan-GAL4.1} Lim3[MI03817-TG4.1] (RRID:BDSC_67450), Mi{PT-GFSTF.1}klg[MI02135-GFSTF.1] (RRID:BDSC_ 59787), Mi{PT-GFSTF.2}GluClalpha[MI02890-GFSTF.2] (RRID:BDSC_60533), Mi{PT-GFSTF.0}TfAP-2 [MI04611-GFSTF.0] (RRID:BDSC_61776), Mi{Trojan-GAL4.2}kn[MI15480-TG4.2] (RRID:BDSC_67516) pJFRC12-10XUAS-IVS-myr::GFP in attP2 (RRID:BDSC_32197), pJFRC19-13XLexAop2-IVS-myr::GFP in su(Hw)attP8 (RRID:BDSC_32211), and pJFRC21-10XUAS-IVS-mCD8::RFP in attP18.

The VAChT-FRT-STOP-FRT-HA transgene (TI{TI}VAChT[FRT-STOP-FRT.HA] (RRID:BDSC_76021) described in *Pankova and Borst (2017)* was used to examine VAChT expression in photoreceptor neurons. Flp-recombinase, either sens-FLP (expressed in R8 cells; *Chen et al., 2014*) (fly stock w[*] P {y[+t7.7] w[+mC]=sens-FLPG5.C}attP18; wg[Sp-1]/CyO; sens[Ly-1]/TM6B, Tb[1] (RRID:BDSC_55768) or ey3.5FLP (expressed in all R-cells; *Bazigou et al., 2007*) (fly stock P{w[+mC]=ey3.5 FLP.B}1, y[1] w [*]; CyO/In(2LR)Gla, wg[Gla-1] PPO1[Bc] (RRID:BDSC_35542) was used to induce VAChT stop-cassette excision.

## Histology

Visualization of split-GAL4 driver line expression patterns with pJFRC51-3XUAS-IVS-Syt::smHA in su (Hw)attP1 and pJFRC225-5XUAS-IVS-myr::smFLAG in VK00005 (*Nern et al., 2015*) or, in a few cases, 20XUAS-CsChrimson-mVenus in attP18 (*Klapoetke et al., 2014*) as reporters was performed as described (*Aso et al., 2014*; *Wu et al., 2016*). Detailed protocols are also available online (https:// www.janelia.org/project-team/flylight/protocols under 'IHC - Anti-GFP', 'IHC - Polarity Sequential' and 'DPX mounting'). Multicolor Flp-out (MCFO) markers were detected by immunolabeling with antibodies against HA, FLAG and V5 epitopes as described (*Nern et al., 2015*). Detailed protocols are also available online (https://www.janelia.org/project-team/flylight/protocols under 'IHC - MCFO'.

For other experiments, brains of female flies were dissected in insect cell culture medium (Schneider's Insect Medium, Sigma Aldrich, #S0146) and fixed with 2% PFA (w/v) (prepared from a 20% stock solution, Electron Microscopy Sciences: 15713) also in cell culture medium for 1 hr at room temperature. Brains were washed with 0.5% (v/v) TX-100 (Sigma Aldrich: X100) in PBS and incubated in PBT-NGS (5% Goat Serum [ThermoFisher: 16210–064] in PBT) for at least 30 min. Incubations with primary antibodies and subsequently, after additional PBT washes, secondary antibodies, were in PBT-NGS at 4℃ overnight. After additional washes with PBT and then PBS, brains were mounted in SlowFadeGold (ThermoFisher: S36937) and imaged on a Zeiss LSM 710 confocal microscope using 20 × 0.8 NA, 40x NA 1.3 or 63 × 1.4 NA objectives. A few specimens were mounted in DPX following the protocol described in *Nern et al. (2015)*. For experiments using only native fluorescence, brains were fixed as above and mounted and imaged after the initial post-fixation washes.

Primary antibodies used in each experiment are indicated in *Supplementary file 1E*. Primary antibodies were anti-GFP rabbit polyclonal (ThermoFisher: A-11122, RRID:AB_221569; used at 1:1000 dilution), anti-GFP mouse monoclonal 3E6 (ThermoFisher: A-11120, RRID:AB_221568; dilution 1:100), anti-dsRed rabbit polyclonal (Clontech Laboratories, Inc.: 632496, RRID:AB_10013483; dilution 1:1000), anti-HA rabbit monoclonal C29F4 (Cell Signaling Technologies: 3724S, RRID:AB_ 1549585; dilution 1:300), anti-FLAG rat monoclonal (DYKDDDDK Epitope Tag Antibody [L5], Novus Biologicals: NBP1-06712, RRID:AB_1625981; 1:200), DyLight 549 or DyLight 550 conjugated anti-V5 mouse monoclonals (AbD Serotec: MCA1360D549GA or MCA1360D550GA, RRID:AB_10850329 or RRID:AB_2687576; 1:500 dilution), anti-cockroach allatostatin (Ast7) mouse monoclonal 5F10 (*Stay et al., 1992*) (also detects *Drosophila* AstA (*Hergarden et al., 2012*); Developmental Studies Hybridoma Bank (DSHB): RRID:AB_528076; dilution 1:5), anti-CadN rat monoclonal DN-Ex #8 (DSHB: RRID:AB_528121; dilution 1:20) (*Iwai et al., 1997*), anti-chaoptin mouse monoclonal 24B10 (DSHB: RRID:AB_528161, dilution 1:20. *Fujita et al., 1982*) and anti-Brp mouse monoclonal nc82 (*Wagh et al., 2006*) (DSHB: RRID:AB_2314866; dilution 1:30).

Secondary antibodies (all from Jackson ImmunoResearch Laboratories, Inc) were DyLight 488-AffiniPure Donkey Anti-Mouse IgG (H+L): 715-485-151, 1:500 dilution; DyLight 594 AffiniPure Donkey anti Rabbit IgG (H+L): 711-515-152, 1:300 dilution; Alexa Fluor 647 AffiniPure Donkey Anti-Rat IgG (H+L): 712-605-153, 1:300 dilution; Alexa Fluor 594 AffiniPure Donkey Anti-Mouse IgG (H+L): 715-585-151,1:300 dilution; Alexa Fluor 647 AffiniPure Donkey Anti-Mouse IgG (H+L): 715-605-151, 1:300 dilution and Alexa Fluor 488 AffiniPure Donkey Anti-Rabbit IgG (H+L): 711-545-152, 1:1000 dilution.

## Image processing

Image analyses and processing were mainly done using Fiji (http://fiji.sc) and Vaa3D (*Peng et al., 2010*). Brightness and contrast were adjusted separately for individual images and channels. Figure panels were assembled using Adobe Indesign. This included selection of fields of view and adjustments of image size. Some images were rotated or mirrored. In some panels with rotated images, empty space outside the original image was filled in with zero pixels. Most of the images in *Figure 1—figure supplement 1C,C'* and *Figure 1—figure supplement 2* show resampled views that were generated from three dimensional image stacks using the Neuronannotator mode of Vaa3D and exported as TIFF format screenshots.

## INTACT purification of nuclei

Frozen adult flies were decapitated by vigorous vortexing. Heads or wings/appendages were then collected on cooled metal sieves (H and C Sieving Systems: 1296, 1297, 1298, 1301). Both flies and purified frozen material can be stored indefinitely at −80°C. In a typical experiment 100–500 frozen heads were added to 5 ml of 20 mM β-glycerophosphate pH7, 200 mM NaCl, 2 mM EDTA, 0.5% NP40, 0.5 mM spermidine, 0.15 mM spermine, 1 mM DTT, 1X complete protease inhibitor (Sigma: 5056489001), 1.5 mg/ml BSA (ThermoFisher: AM2618), 1 mg/ml torula yeast RNA (ThermoFisher: AM7118), 0.6 mg/ml carboxyl coated Dynabeads (ThermoFisher: 14306D) and 2μg anti-GFP antibody (ThermoFisher: G10362, RRID:AB_2536526). Homogenization was carried out on ice by 50 tractions in a Dounce homogenizer using the tight pestle followed by filtration over a 10μm cup filter (Partec: 0400422314). Released chromatin and broken nuclei were adsorbed to carboxyl coated magnetic beads for 30 min at 4°C with constant rotation. Beads were removed on a magnetic stand and the supernatant was diluted to 50 ml with 20 mM β-glycerophosphate pH7, 200 mM NaCl, 2 mM EDTA, 0.5% NP40, 0.5 mM spermidine, 0.15 mM spermine, 1 mM DTT and 1X complete protease inhibitor (Sigma: 5056489001), filtered over a 1μm cup filter (Pluriselect: 435000103) and split into two equal volumes. A 40% Optiprep (Sigma: D1556), 20 mM β-glycerophosphate pH7, 2 mM EDTA and 0.5% NP40 solution was then gently placed under each aliquot, followed by a lower layer of 50% Optiprep, 20 mM β-glycerophosphate pH7, 2 mM EDTA and 0.5% NP40. Nuclei were then pelleted on to the 50% layer for 30 min at 2300Xg. Purified nuclei were passed over a 10μm cup filter, diluted to 10 ml with 20 mM β-glycerophosphate pH7, 200 mM NaCl, 2 mM EDTA, 0.5% NP40, 0.5 mM spermidine, 0.15 mM spermine, 1 mM DTT and 1X complete protease inhibitor and incubated with 30μl of protein G Dynabeads (ThermoFisher: 10004D) for 40 min on ice with occasional agitation. Bead-bound nuclei were recovered on a magnet stand followed by a 20 min incubation on ice in 9mls of 20 mM β-glycerophosphate pH7, 300 mM NaCl, 1M urea, 0.5% NP40, 2 mM EDTA, 0.5 mM spermidine, 0.15 mM spermine, 1 mM DTT, 1X complete protease inhibitor, 0.075 mg/ml torula RNA and 0.05 U/ml Superasin (ThermoFisher: AM2696). Nuclei were then recovered on a magnet stand, resuspended in 1 ml of the previous buffer, passed over a 10μm cup filter, a 5μl aliquot was withdrawn for quantitation and the remainder of the sample solubilized in Arcturus Picopure RNA extraction buffer (ThermoFisher: KIT0204).

## TAPIN purification of nuclei

100–3000 frozen heads were added to 5 ml of 20 mM sodium acetate pH8.5, 2.5 mM MgCl$_2$, 250 mM sucrose, 0.5% NP-40, 0.6 mM spermidine, 0.2 mM spermine, 1 mM DTT, 1X complete protease inhibitor, 0.5 mg/ml torula RNA, 0.6 mg/ml carboxyl coated Dynabeads and 2μg anti-GFP antibody (*Supplementary file 1F*). Homogenization was carried out on ice by 50 tractions in a Dounce homogenizer using the tight pestle followed by filtration over either a 10 or 20μm cup filter (Partec: 0400422314 or 040042315). Released chromatin and broken nuclei were adsorbed to carboxyl

coated magnetic beads for 30 min at 4°C with constant rotation. Unbound antibody was removed by incubating the sample on ice for 20 min with 100μl of UNOsphere SUPra resin (Biorad: 1560218), which was previously washed 2X with 500 mM sodium acetate ph8.5/0.5% NP40 and 2 × 20 mM sodium acetate ph8.5/0.5% NP40. After the resin was removed on a 10μm cup filter and the carboxyl beads on a magnet stand, the nuclei-containing supernatant was mixed with an equal volume of 500 mM sodium acetate pH8.5, 250 mM sucrose, 6 mM EGTA, 6 mM EDTA, 0.6 mM spermidine, 0.2 mM spermine, 1 mM DTT, 1X complete protease inhibitor, 0.25 mg/ml torula yeast RNA and 30μl Protein A Dynabeads (ThermoFisher: 10002D) (*Supplementary file 1F*). A 2 hr incubation on ice with occasional agitation was used to recover tagged nuclei. Bead-bound nuclei were then recovered on a magnet stand and washed twice with 250 mM sodium acetate ph8.5, 250 mM sucrose and 0.1% NP40 (*Supplementary file 1F*). Nuclei were then released at 37°C for 1 hr by incubation in 50μl of 10 mM Tris pH7.5, 2.5 mM $MgCl_2$, 0.5 mM $CaCl_2$, 250 mM sucrose, 0.1% NP40, 1 mg/ml torula RNA, 40 units RNAsin (Promega: N2515), 2 units DNAseI (NEB: M0303L), 320 units IdeZ protease (NEB: P0770S) (*Supplementary file 1F*). The sample was diluted to 100μl with 10 mM Tris pH7.5, 2.5 mM $MgCl_2$, 0.5 mM $CaCl_2$, 250 mM sucrose and 0.1% NP40, EGTA was added to 1 mM and the suspension was rapidly triturated 100 times. After returning the sample to a magnet stand, 90μls of buffer containing released nuclei was removed and added to 1.5μl of Protein G Dynabeads that were previously resuspended in 10μl of 10 mM Tris pH7.5, 2.5 mM $MgCl_2$, 0.5 mM $CaCl_2$, 250 mM sucrose and 0.1% NP40. The second binding reaction was run for 1–3 hr on ice with occasional agitation, followed by two 250μl washes in 10 mM Tris pH7.5, 2.5 mM $MgCl_2$, 0.5 mM $CaCl_2$, 250 mM sucrose and 0.1% NP40. Prior to the last wash a 5μl aliquot was removed for quantitation and the remainder of the sample was solubilized in Arcturus Picopure RNA extraction buffer.

## RNA-seq library construction

Nuclear RNA was DNAseI (Qiagen: 79254) treated and purified using the Arcturus PicoPure (ThermoFisher: KIT0204) system as instructed by the supplier. Purified RNA was mixed with a 1:100,000 dilution of ERCC standard RNA mix #1 (ThermoFisher: 4456740) and amplified using the Nugen Ovation v2 system (Nugen: 7102–32). cDNA was then blunted, ligated to barcoded linkers (Nugen: 0319–32, 0320–32) and sequenced on an Illumina Hiseq 2500 to 50 bp read length using Rapid Run flow cells.

In total we built 266 RNA-seq libraries, including 46 INTACT-seq, 196 TAPIN-seq, eight total RNA libraries from dissected tissues, and 16 control libraries that we used to characterize each INTACT/TAPIN-seq step (*Figure 2C*, *Supplementary file 1A*).

## RNA-seq data processing

We trimmed five nucleotides from the 5′ end of reads using seqtk (https://github.com/lh3/seqtk) to remove potential contaminating adapter sequence from the NuGen Ovation kit. We estimated the abundance of annotated genes using kallisto (v0.43.1; *Bray et al., 2016*) to pseudo-align trimmed reads to the fly transcriptome (cDNA and ncRNA transcript sequences from ENSEMBL release 91, based on FlyBase release 2017_04), ERCC spike-ins, and the INTACT construct sequences GAL4-DBD, p65-AD, and UNC84_2XGFP. ERCC, INTACT tag constructs, and rRNA genes were removed from the abundance tables and the estimated abundances of the remaining genes were renormalized to one million total transcripts. The ERCC spike-ins and nuclear yield values allowed us to convert relative transcript abundance (in Transcripts Per Million, TPM) to absolute abundance (*Figure 3—figure supplement 1G*). However, we only used relative abundance for our analyses. We also aligned the trimmed reads to the genome using STAR (v2.5.3c; *Dobin et al., 2013*) and evaluated gene body coverage bias using Picard (v 1.9.1; http://broadinstitute.github.io/picard).

We used three criteria to quantify the quality of each library: the number of genes detected, the pearson correlation between transcript abundances measured in replicates, and the cDNA yield. We used only high-quality libraries (at least 8500 genes detected, 3μg cDNA yield, and 0.85 Pearson's correlation of transcript abundances in two biological replicates) as input to the model described below.

## Comparison to published single cell and FACS-seq datasets

We obtained genes reported to mark the single cell clusters in a recent scRNA-seq study of the optic lobe (*Konstantinides et al., 2018*). We also obtained the cluster assignments for each single cell in this dataset from the SCope database (*Davie et al., 2018*), using Seurat clustering resolution 4.0, as reported by the authors. We analyzed FACS-sorted RNA-seq samples reported by Konstantinides et al. by downloading the raw sequencing reads from the NCBI Sequence Read Archive (https://www.ncbi.nlm.nih.gov/sra) and estimating transcript abundance using kallisto and the same transcriptome index as above.

To compare actual and predicted cluster sizes, we used the following numbers for cells per type: Cell types that are thought to be present once (L1, L2, L3, L4, L5, T1, Mi1, T2, T3, Tm2, Tm9, C2,C3, T4a,T4b,T4c,T4d,T5a,T5b,T5c,T5d) or approximately once (Dm8, Tm3) per medulla column based on EM studies (*Takemura et al., 2013*; *Takemura et al., 2015*; *Takemura et al., 2017*) and light microscopy of specific driver lines (for example, T4/T5 *Mauss et al., 2014*; lamina cells *Tuthill et al., 2013*, T4 inputs *Strother et al., 2017*, Dm8 *Nern et al., 2015*) were estimated as 1 cell/column * 750 columns/medulla * two hemispheres = 1500 cells per brain. Estimates for Dm12 (~120×2 cells per brain, *Nern et al., 2015*) and Lawf2 (~140×2 cells per brain, *Tuthill et al., 2014*) were as published. We performed new counts for Pm3 (mean +/- SD 37 +/- 3 cells per optic lobe; n = 4 optic lobes; driver line SS00328) and Lawf1 (151+/- 7 cells per optic lobe; n = 4 optic lobes; two optic lobes each for driver lines SS00689 and SS00800). No precise count was available for Tm5c; since this cell type is known to be present in many but not all medulla columns, we used an estimate of 400 cells per optic lobe x two hemispheres = 800 cells per brain (*Takemura et al., 2013*; *Melnattur et al., 2014*; *Karuppudurai et al., 2014*).

To compare our bulk TAPIN-seq profiles to published single cell datasets, we used non-negative least squares regression to model each bulk TAPIN-seq profile as a linear weighted sum of single cell clusters and a profile-specific residual (TAPIN ~cluster). We began with single-cell expression matrices extracted from the loom files deposited in the SCope database, using the SCopeLoomR package (*Davie et al., 2018*). After normalizing the transcriptome of each cell from transcript counts to counts per million, we calculated the mean transcriptome for each single cell cluster. We then selected genes that were enriched in either single cell clusters or TAPIN-seq cell types, using the following criteria (adapted from *Cao et al., 2019*): union of the top-50 genes that were most enriched relative to the average of all other clusters (or TAPIN-seq cell types) and the top-50 genes that were most enriched relative to the maximum level in all other clusters (or TAPIN-seq cell types). We then performed NNLS regression using the Lawson-Hanson implementation (*Lawson and Hanson, 1974*) available through the nnls R package (*Mullen and Stokkum, 2012*). To visualize the results, we created heatmaps of the regression coefficients, normalizing the values for each single cell cluster across TAPIN-seq cell types.

We also performed the NNLS regression in the opposite direction (cluster ~TAPIN, explaining single cell clusters as combination of TAPIN-seq profiles), as we thought this direction would more naturally describe mixed clusters composed of multiple cell types (e.g., all photoreceptors, or all monopolar cells). In practice however, this regression assigned coefficients of exactly zero to several TAPIN-seq profiles – as expected given the power of NNLS to recover sparse solutions (*Slawski and Hein, 2013*) and the collinearity amongst TAPIN-seq profiles (e.g., highly correlated expression amongst photoreceptor subtypes). In contrast, the TAPIN ~ cluster regression correctly matched mixed clusters with the corresponding TAPIN-seq profiles.

## Inferring expression state from transcript abundance

We begin with a catalog of $S$ RNA-seq samples generated from nuclei isolated from cell type *cell(s)* and the estimated abundance (in TPM), of $E_{gs}$ transcripts from gene $g$ in each sample $s$. We consider only protein-coding genes with at least 10 TPM abundance in at least one sample (n = 12,377 of 13,931 total coding genes).

To interpret $E_{gs}$, we assume that all genes express in either an 'on' or an 'off' state. Our goal is to infer from these abundances the probability that each gene is expressed in each cell type, $P(z_{gc} = \mathrm{on})$. Depending on the cell types in our catalog, we will observe some genes in both on and off states (bimodal), while others are exclusively off (unimodal-off) or on (unimodal-on). We deal with these scenarios in turn below.

Assuming that a gene is bimodal, we model its expression as arising from a mixture of two gene-specific log-normal distributions describing expression in cells where the gene is off, $P(E_g|z = \mathrm{off})$, and those where the gene is on, $p(E_g|z = \mathrm{on})$, combined with a mixing weight, $\pi_g$. We use the same standard deviation for both on and off distributions to ensure a monotonic relationship between transcript abundance and the posterior probability of the on state. If we use different standard deviations for each component distribution, the wider one would become more probable than the narrower one at both low *and* high expression levels.

$$\log E_g|z \sim \mathcal{N}(\mu_{gz}, \sigma_g)$$

We estimate the posterior probability of the on state (assuming bimodal expression):

$$P(z_g = \mathrm{on}|\mathrm{bimodal}) = \frac{\pi_g p(E_{gs}|z = \mathrm{on})}{\pi_g p(E_{gs}|z = \mathrm{on}) + (1 - \pi_g) p(E_{gs}|z = \mathrm{off})}$$

We treated each replicate sample of the same driver as an independent probe of the same underlying driver-line expression state. To combine replicates of the same driver we sum over their likelihoods:

$$P(z_{gd} = \mathrm{on}|\mathrm{bimodal}) = \frac{\pi_g \Pi_s p(E_{gs}|z = \mathrm{on})}{\pi_g \Pi_s p(E_{gs}|z = \mathrm{on}) + (1 - \pi_g) \Pi_s p(E_{gs}|z = \mathrm{off})}$$

Similarly, to combine samples from the same cell type we sum over their likelihoods:

$$P(z_{gc} = \mathrm{on}|\mathrm{bimodal}) = \frac{\pi_g \Pi_s p(E_{gs}|z = \mathrm{on})}{\pi_g \Pi_s p(E_{gs}|z = \mathrm{on}) + (1 - \pi_g) \Pi_s p(E_{gs}|z = \mathrm{off})}$$

We estimated parameters for each gene-specific mixture model by maximizing the likelihood for the observed sample-level data:

$$\mathcal{L} = \prod_g \prod_s (\pi_g p(E_{gs}|z = \mathrm{on}, \mu_{gz}, \sigma_g) + (1 - \pi_g) p(E_{gs}|z = \mathrm{off}, \mu_{gz}, \sigma_g))$$

Because we assume independence of genes, we separately optimized the model parameters for each gene. To model the possibility that a gene is unimodally expressed across the cell types we analyzed, we also model the data using a single log-normal distribution, estimating the distribution parameters μ and σ and estimating the data likelihood as:

$$\mathcal{L} = \prod_g \prod_s p(E_{gs}|\mu_g, \sigma_g)$$

Deciding whether a gene is bimodally or unimodally expressed is an example of the model selection problem in statistics. To compare the quality of the unimodal and bimodal models for each gene, we used a recently developed approach to leave-one-out cross-validation that uses Pareto-smoothed importance sampling (PSIS-LOO; *Vehtari et al., 2017*). Specifically, we performed 10-fold cross validation, by randomly holding out 1/10 of the samples as a 'test' set (requiring that at least one replicate of each driver exist in the remaining 'training' set), fitting the models using only the training data, and then evaluating the likelihood of the test data using the fitted parameters. Each of the ten cross-validation fits, *i*, returns an ensemble of $S = 500$ draws from the posterior distribution of the model parameters. We estimated the expected log pointwise predictive density (elpd) of each cross-validation fit by evaluating the likelihood of each held-out dataset *i* using each parameter draw *s*:

$$\widehat{elpd_i} = log(\frac{1}{draws} \sum_s^{draws} p(y_i|\theta_{s,k}))$$

We then combined the pointwise log-likelihoods for each cross-validation fit to calculate a single estimate for each model:

$$\widehat{elpd} = \sum \widehat{elpd_i}$$

To compare the unimodal and bimodal models, we calculated the difference in elpd as well as its standard error:

$$\Delta\widehat{elpd} = \widehat{elpd}^{\text{bimodal}} - \widehat{elpd}^{\text{unimodal}}$$

$$\text{se}(\Delta\widehat{elpd}) = \sqrt{nV_i^n(\widehat{elpd}_i^{\text{bimodal}} - \widehat{elpd}_i^{\text{unimodal}})}$$

We then picked the model with the higher elpd, unless the difference in elpd was within two multiples of its standard error ($abs(\Delta\widehat{elpd}) \leq 2 \cdot \text{se}(\Delta\widehat{elpd})$, corresponding approximately to the half-width of a 95% confidence interval in a normal sampling distribution) in which case we considered the two models' performance to be indistinguishable and chose the simpler unimodal model.

If we decide a gene is unimodal, we must still decide if it is expressed or not. To model the expression state of unimodal genes, we created two separate log-normal distributions of abundances of confidently bimodal genes ($\Delta\widehat{elpd}>10$) using samples where they were either estimated to be on according to the bimodal model ($P(z_{gs} = \text{on}|\text{bimodal})>0.9$) and where they were estimated to be off ($P(z_{gs} = \text{on}|\text{bimodal})<0.1$), combined with a mixing weight, $\pi$, set to the fraction of datapoints that were estimated to be 'on' according to the bimodal model.

$$\log E_g|z \sim \mathcal{N}(\mu_z, \sigma_z)$$

We estimate the posterior probability of the on state assuming unimodal expression as:

$$P(z_g = \text{on}|\text{unimodal}, \mu_g) = \frac{\pi p(\mu_g|z = \text{on})}{\pi p(\mu_g|z = \text{on}) + (1 - \pi)p(\mu_g|z = \text{off})}$$

To build the final matrix of $P(z_{gs} = \text{on})$ calls, we used bimodal estimates for genes where the bimodal model was a better fit than the unimodal model, and the unimodal estimates for the remaining genes.

$$P(z_{gs} = \text{on}) = \begin{cases} P(z_{gs} = \text{on}|\text{bimodal}), & \text{if } \widehat{elpd}\text{bimodal}(g) > \widehat{elpd}\text{unimodal}(g) \text{ and } \Delta\widehat{elpd} > 2 \cdot \text{se}(\Delta\widehat{elpd}), \\ P(z_g = \text{on}|\text{unimodal}), & \text{otherwise} \end{cases}$$

We did not include the transcriptomes of the dissected samples in the mixture models because we were concerned that their cellular heterogeneity would violate our assumption of binary gene expression in each sample. That is, genes expressed in a subset of the cells of a dissected sample would give rise to transcript abundance intermediate between the off and on states, and thus make it more difficult to accurately infer the component distributions. However, in some cases the dissected samples could be useful for interpreting transcript levels in the cells that we profiled, by providing examples that extend the observed dynamic range. For example, in the case of a gene expressed in a dissected tissue, but not in the cells that we specifically profiled, the dissected levels would add 'on' examples that would make it easier to interpret the levels in the cell types as 'off'. To use the dissected samples to better model dynamic range, we added two 'dummy' samples to each model: the minimum and maximum observed level across both the cell catalog and the dissected samples. This choice allowed us to use the dissected levels if they in fact outflanked the cell type-specific levels, while not confusing the model with intermediate abundance levels. Once the models were fit, we could use the inferred parameters to estimate expression probabilities for samples that were not used in the model fit. For example, we estimated the probabilities of expression in the dissected samples to search for genes expressed exclusively in the dissected samples and not in the anatomically defined cell type libraries, indicating potential markers for cells that we did not specifically profile.

We implemented all models using RStan (**Stan Development Team, 2017**; **Carpenter et al., 2017**) to infer the posterior distribution of unknown parameters using hamiltonian Markov chain Monte Carlo. We used the same weak prior (N(7,5)) for the mean log-expression levels of both on and off components, allowing us to use Stan's positive_ordered data type to describe the location of the two components. The models were sampled using 4 independent Markov chains, each run for 500 iterations. Additional details of the model fitting procedure can be found in the STAN files specifying the model and the R code that calls the model fitting procedures (github repository http://

github.com/fredpdavis/opticlobe; copy archived at https://github.com/elifesciences-publications/opticlobe; *David, 2020*).

## Evaluating model accuracy

To evaluate the accuracy of the mixture modeling approach we created a benchmark set of expression data extracted from FlyBase. Specifically, we queried the FlyBase website (http://flybase.org) for genes expressed in the optic lobe or the photoreceptor. The resulting benchmark set included 193 positive and 4 negative expression datapoints. We quantified the model's accuracy on this benchmark in two ways. First, we quantified concordance between the benchmark expression state and our model's inferred state. Second, we computed the cumulative distribution function of the inferred probabilities of expression for the positive benchmark datapoints.

## Expression-based tree of cell types

To study cell relationships, we used phylogenetic tree-building to compare their expression profiles. We first selected a subset of genes with on-component means of at least $\exp(3) \sim 21$ TPM and difference between on and off components of at least $\exp(1.5) \sim 4.5$ fold. We then encoded the expression profile of each cell as a 'sequence' of expression states, where each position represents a gene, and the character indicates the gene is expressed ('A', $P(z_{gc} = \text{on}) > 0.8$), not expressed ('C', $P(z_{gc} = \text{on}) < 0.2$), or its expression is uncertain ('N', $0.2 < P(z_{gc} = \text{on}) < 0.8$). We computed the Hamming distance between pairs of expression 'sequences' considering only unambiguous positions, using the dist.dna() routine in the ape R package (*Paradis et al., 2004*). We then used the minimum evolution approach to estimate the 'phylogeny' of the cells, using the balanced weighting scheme (*Desper and Gascuel, 2002*), as implemented in the ape fastme.bal() routine. We then built trees from 1000 bootstrapped replicates and quantified the support for each branch on the original tree. We visualized the tree using the phytools R package (*Revell, 2012*).

## Identifying marker genes

We identified marker genes specifically enriched in individual cell types and groups of cells (photoreceptor, glia, muscle, neuron) by searching for genes inferred to be almost exclusively expressed in a single cell type or cell group ($P(z_{gc} = \text{on}) >= 0.9$ for all cells within a group, and at most two cells outside a group) and with transcript abundance higher than all cells outside the group.

## Evaluating expression patterns for genes with different functions

We used FlyBase Gene Groups (release 2018_02) to assign functions to genes, and considered the most terminal groups in the hierarchy that had at least 10 genes.

## Mapping receptor expression onto synapses

To map receptor expression onto synaptic connectivity, we first obtained synapse pairs from *Takemura et al. (2013)* to identify synaptic targets of R8 (cell #111), R7 (cell #205), and C2 (cell #214) cells in the medulla. When multiple instances of a cell type were available in the synaptic table, we chose the one with the greatest number of synaptic partners. For target cell types that we profiled with TAPIN/INTACT-seq, we discretized their expression as either on (p(on)>=0.8) or off (p(on) <0.8). For cell types that we did not profile, we classified them as unknown receptor expression.

## Data and software availability

All raw and processed transcriptome data is available from NCBI GEO (accession GSE116969). The shell scripts used to process the raw RNA-seq data, and the R and Stan programs that implement the mixture model as well as generate all figures and tables in this paper are available at github (http://github.com/fredpdavis/opticlobe). The cell type-level expression table can be explored interactively at http://www.opticlobe.com.

## Acknowledgements

This project was supported by the Howard Hughes Medical Institute. FPD was supported in part by the Intramural Research Program of the National Institute of Arthritis and Musculoskeletal and Skin

Diseases of the National Institutes of Health. We would like to thank Teresa Tian and all members of the Janelia Fly Facility for their help in maintaining and rearing flies, Tanya Wolff and Arnim Jenett for the central complex and Heather Dionne for the l-LNv split-GAL4 lines, and the FlyLight Project Team for imaging of split-GAL4 expression patterns.

## Additional information

### Funding

| Funder | Grant reference number | Author |
|---|---|---|
| National Institute of Arthritis and Musculoskeletal and Skin Diseases | Intramural Research Program | Fred P Davis |
| Howard Hughes Medical Institute | | Fred P Davis<br>Aljoscha Nern<br>Serge Picard<br>Michael B Reiser<br>Gerald M Rubin<br>Sean R Eddy |

The funders had no role in study design, data collection and interpretation, or the decision to submit the work for publication.

### Author contributions

Fred P Davis, Conceptualization, Data curation, Software, Formal analysis, Investigation, Methodology, Writing - original draft, Writing - review and editing; Aljoscha Nern, Conceptualization, Data curation, Validation, Investigation, Methodology, Writing - original draft, Writing - review and editing; Serge Picard, Resources, Investigation; Michael B Reiser, Gerald M Rubin, Funding acquisition, Writing - review and editing; Sean R Eddy, Conceptualization, Supervision, Funding acquisition, Writing - review and editing; Gilbert L Henry, Conceptualization, Data curation, Investigation, Methodology, Writing - original draft, Writing - review and editing

### Author ORCIDs

Fred P Davis https://orcid.org/0000-0001-8294-1610
Aljoscha Nern https://orcid.org/0000-0002-3822-489X
Michael B Reiser http://orcid.org/0000-0002-4108-4517
Gerald M Rubin http://orcid.org/0000-0001-8762-8703
Sean R Eddy http://orcid.org/0000-0001-6676-4706
Gilbert L Henry https://orcid.org/0000-0003-1590-323X

### Decision letter and Author response

Decision letter https://doi.org/10.7554/eLife.50901.sa1
Author response https://doi.org/10.7554/eLife.50901.sa2

## Additional files

### Supplementary files

• Supplementary file 1. Supplemental tables. (**A**) All drivers. (**B**) RNAseq samples. (**C**) Benchmark entries. (**D**) Marker genes. (**E**) Anatomy details. (**F**) TAPIN-seq buffers.

• Supplementary file 2. Key Resource Table. List of key resources and reagents used in this study.

• Transparent reporting form

### Data availability

All raw and processed transcriptome data is available from NCBI GEO (accession GSE116969).

The following dataset was generated:

| Author(s) | Year | Dataset title | Dataset URL | Database and Identifier |
|---|---|---|---|---|
| Davis FP, Nern A, Picard S, Reiser MB, Rubin GM, Eddy SR, Henry GL | 2019 | A genetic, genomic, and computational resource for exploring neural circuit function | https://www.ncbi.nlm.nih.gov/geo/query/acc.cgi?acc=GSE116969 | NCBI Gene Expression Omnibus, GSE116969 |

The following previously published datasets were used:

| Author(s) | Year | Dataset title | Dataset URL | Database and Identifier |
|---|---|---|---|---|
| Konstantinides N, Kapuralin K, Desplan C | 2018 | RNA sequencing of *Drosophila melanogaster* optic lobe cell types | https://www.ncbi.nlm.nih.gov/geo/query/acc.cgi?acc=GSE103772 | NCBI Gene Expression Omnibus, GSE103772 |
| Konstantinides N, Kapuralin K, Desplan C | 2018 | Single-cell RNA sequencing of *Drosophila melanogaster* optic lobe cells | https://www.ncbi.nlm.nih.gov/geo/query/acc.cgi?acc=GSE103771 | NCBI Gene Expression Omnibus, GSE103771 |
| Davie K, Janssens J, Koldere D, Aerts S | 2018 | A single-cell transcriptome atlas of the ageing *Drosophila* brain | https://www.ncbi.nlm.nih.gov/geo/query/acc.cgi?acc=GSE107451 | NCBI Gene Expression Omnibus, GSE107451 |

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
