## [Decision Letter]

Thank you for submitting your article "A genetic, genomic, and computational resource for exploring neural circuit function" for consideration by *eLife*. Your article has been reviewed by three peer reviewers, and the evaluation has been overseen by a Reviewing Editor and K VijayRaghavan as the Senior Editor. The reviewers have opted to remain anonymous.

The reviewers have discussed the reviews with one another and the Reviewing Editor has drafted this decision to help you prepare a revised submission.

The three reviewers are in agreement that this work is a very valuable contribution to the literature and that it should be published in *eLife*. I am also of the same opinion. However, they all three raised issue that will require some textual changes, better explanations of the modeling. Here is a summary of the most important issues.

Summary

In this manuscript Davis et al. have used different enhancer combinations to mark diverse subtypes of cells, mostly from the visual system, and conduct RNA-seq analysis on isolated nuclei to assign gene expression patterns observed in 67 cell types. The authors first refined the INTACT method, which helps to isolate nuclei specified by a GAL4 expression pattern. In the new method (TAPIN) the authors add tandem affinity purification to the nucleus isolation pipeline. The authors show that this modification increases sensitivity of detection of transcripts and specificity of isolated nuclei compared to the previous methodology. The authors convert the often-graded expression levels to two normal distributions of binary on-off states. They then convert this data to probability of expression scores by using mathematical modeling. The accuracy of this modelling approach is demonstrated in multiple examples. The authors compare the expression patterns that they obtained to the recently available single cell sequencing data sets. Interestingly very few expression patterns directly map to a single cell transcription profile. Nevertheless, this comparison helped to determine the identity of many single cell sequencing clusters obtained in previous studies. Finally, the authors analyze the expression patterns of neurotransmitters and their receptors. By combining high resolution connectome obtained by EM with the expression patterns obtained by TAPIN and INTACT methods the authors find multiple wiring paradigms in the *Drosophila* brain.

The authors try to find relationships between cell types. They unsurprisingly find grouping of cell types with similar developmental origins and structures. They compare their results to published single-cell sequencing datasets from the optic lobes and the brain and confirmed that the single-cell datasets consisted of clusters that include more than one cell types, which was already known based on the number of clusters itself. They manage to annotate and correct some of the single-cell clusters, showing that bulk transcriptomes can be used to interpret single-cell clusters. They go on to analyze neurotransmitter and neurotransmitter receptor expression in different cell types, before finally trying to interpret connectomic data in the light of their transcriptomes.

Major comments:

– There are issues that need to be discussed more openly. For instance, in many instances the RNA expression pattern of a cell type is referred to a cells expression pattern. This would assume that the expression pattern is homogenous in a given cell type. This is a big assumption. The fact that the TAPIN expression patterns do not directly correlate with the single cell expression profiles can indicate that the cells from a cell type can have divergence in their expression profile. This has implications about the cell types that show multiple neurotransmitters. This issue should be more clearly discussed.

– Another issue is that the authors use the RNA levels and protein levels interchangeably, disregarding the possibility of post transcriptional regulation (This is more pronounced when they use the RNA expression data about cell adhesion molecules in Figure 4—figure supplement 1). This should also be addressed.

– The extent to which single cell sequencing clusters can be compared to TAPIN profiles should be better discussed. The TAPIN profiles are RNA expression of cells that share one or two enhancers (by the use of GAL4 or splitGAL4). Unbiased clustering of single cell sequencing data is based on multiple principle components. The manuscript compares the two approaches and suggest one is better than the other. These are complementary approaches and the data are different and difficult to compare (due to possible RNA expression heterogeneity within cell types). This is reflected in the fact that mid-level hierarchical clustering was not well supported by the TAPIN data (Figure 4A) whereas discrete clustering can be observed in single cell sequencing data. The best experiment to address this would be to do single cell sequencing on TAPIN isolated population of neurons nuclei for one or two cell types. I am not requiring this but they should discuss this better

– Although this is a resource paper, the resource is not accessible to people with limited programming or computational skills. In the web portal opticlobe.com, a simple heatmap can be generated based on the user input. This visualization can hardly reflect the depth and complexity of the data. Please refer to https://gtexportal.org/home/ as a reference to add more tools for general users.

– The description of the mixture model is confusing. The lowercase and uppercase p is mixed for different probabilistic events. Do they denote different functions? Conditioned on "bimodal", the posterior probability equation in subsection “Inferring expression state from transcript abundance” is very confusing. Did you integrate out the p(bimodal)? If not, how is the Bayesian rule applied here? The math equations are not well defined.

– On the model selection section, the first equation used sums of probability, the second equation uses a log sum of probability. Is there a reference or proof for such modeling choices from STAT or machine learning literature?

– The mixture model is well studied in both Bayesian and Frequentist framework, how does this approach compare to the standard Bayesian Gaussian mixture model?

– Figure 3C is the main result of the mixture model and the lower heatmap showed about HALF of the cells in the matrix are read and the other half is blue across all the genes and samples. This is likely due to an artifact of the customized mixture modeling. Since most of the results in the manuscript depend on the probability estimated from the mixture model, it is crucial to show mathematically that the model is correct.

---

## [Author Response]

Major comments:– There are issues that need to be discussed more openly. For instance, in many instances the RNA expression pattern of a cell type is referred to a cells expression pattern. This would assume that the expression pattern is homogenous in a given cell type. This is a big assumption. The fact that the TAPIN expression patterns do not directly correlate with the single cell expression profiles can indicate that the cells from a cell type can have divergence in their expression profile. This has implications about the cell types that show multiple neurotransmitters. This issue should be more clearly discussed.

The reviewers raise several important points. By construction, our approach aims to profile the bulk expression of all the cells in a given GAL4 or split-GAL4 pattern. Most of the driver lines we used were selected to be primarily expressed in cells with shared anatomical properties that distinguish these cells from the cells of other types. In particular, many of the optic lobe cell types we profile are anatomically very well defined and have long been accepted as distinct cell types in the literature on the fly visual system. While we go through extensive efforts to characterize the specificity of the driver patterns by anatomy, it is true that the cells in such a pattern could exhibit different profiles. In a few cases such heterogeneity is expected: As we discuss in the text and detail in Supplementary file 1A, some of our driver lines combine related cell types or include “contaminants” (cells of other types) in addition to the targeted neurons. In addition, as we now explicitly discuss, cells without obvious anatomical differences at the light microscopy level could still include hidden subtypes with distinct gene expression (such as the pale and yellow subtypes of R7 and R8 photoreceptor neurons with different rhodopsin expression but similar anatomy). Finally, there are of course sources of variation in gene expression other than cell type differences. For example, even within the exact same cell, the transcriptome can vary over time, e.g. due to circadian rhythms or, over a different time-scale, aging (Davie et al., 2018).

While we sometimes refer to a cell type’s expression as a cell’s expression, we now clarify this wording in the text: “(For simplicity, we will hereon refer to cell type as just cell. This is not meant to exclude the possibility of heterogeneity within the individual cells of a profiled population.)”

For many applications, combined data for populations of related cell types can be useful, since such cells are likely to share many molecular properties. For example, we profile driver lines expressed in motion-sensitive T4/T5 neurons that combine eight anatomically distinct subpopulations. As we highlight in Figure 4D-H, using drivers that express in different subsets of these eight cell types, we can identify clear differences between the transcriptomes of some subtypes. Nevertheless, most molecular properties of T4/T5 cells appear to be shared across the different subpopulations. Recently, the Zipursky lab has sorted T4 and T5 cells and used single cell RNA-seq to more comprehensively characterize T4/T5 subtypes (Kurmangaliyev et al., *eLife* 2019). This scRNA analysis identified eight T4/T5 subclusters, exactly in line with the predictions from anatomy. This work also confirms our finding of Tfap2 and klg as markers for different T4/T5 subsets and supports the general similarity of the eight subtypes.

To the reviewers specific point of how heterogeneity within cell types might explain differences between TAPIN and single cell profiles, we would remind the reviewer that we compared our TAPIN profiles to the average profile of single cells within each cluster, not to individual single cells. And so, any heterogeneity within a single cell cluster would be averaged out in the comparison to TAPIN-seq profiles. In fact, the resolution of the single cell maps that we analyzed (e.g. collapsing the eight T4/T5 subtypes into a single cluster) makes it unlikely that they resolved many named cell types into previously unknown distinct molecular subtypes, and the authors of those maps make no such claims.

Finally, we agree that, using only a given bulk profile, it is not possible to formally rule out that two neurotransmitter markers (or, for that matter, any other combination of transcripts) that appear co-expressed are actually present in distinct, non-overlapping sets of cells. Single cell studies do not necessarily avoid this issue either: For example, Konstatinides et al. report two clusters with apparent co-expression of cholinergic with glutamatergic or GABAergic markers, respectively, but are unable to determine whether this is true co-expression (and in addition do not know the identity of the cell type(s) involved). In our view, the best approach to resolve such uncertainty are follow-up experiments with an independent method: For example, we used immunolabeling of a tagged VAChT protein (Figure 8A) to confirm the unexpected expression of cholinergic markers in R8 photoreceptors cells. Together with published work on histamine expression in R-cells and our observation that histamine receptors are expressed in some but not all synaptic partners of R8 cells (Figure 8B-F), this strongly suggests a co-transmitter phenotype of these neurons. In general, because TAPIN-seq profiles are linked to markers (driver lines) for the profiled cells, performing validation experiments for selected transcripts is straightforward and we provide several examples of this.

– Another issue is that the authors use the RNA levels and protein levels interchangeably, disregarding the possibility of post transcriptional regulation (This is more pronounced when they use the RNA expression data about cell adhesion molecules in Figure 4—figure supplement 1). This should also be addressed.

All of the TAPIN-seq data that we measured were at the transcript level. We compared it to protein expression as a more stringent functional-requirement on “real” expression. The reviewer is correct that there could be additional levels of post-transcriptional regulation and we clarify this point in the text.

“Protein expression can of course differ from that of the corresponding mRNA due to post-transcriptional regulation. However, since most functional interpretations of transcriptome data are implicitly about protein expression, we used this as a more stringent and practical test of our model.”

– The extent to which single cell sequencing clusters can be compared to TAPIN profiles should be better discussed. The TAPIN profiles are RNA expression of cells that share one or two enhancers (by the use of GAL4 or splitGAL4). Unbiased clustering of single cell sequencing data is based on multiple principle components. The manuscript compares the two approaches and suggest one is better than the other. These are complementary approaches and the data are different and difficult to compare (due to possible RNA expression heterogeneity within cell types). This is reflected in the fact that mid-level hierarchical clustering was not well supported by the TAPIN data (Figure 4A) whereas discrete clustering can be observed in single cell sequencing data. The best experiment to address this would be to do single cell sequencing on TAPIN isolated population of neurons nuclei for one or two cell types. I am not requiring this but they should discuss this better

We agree with the reviewer that single cell sequencing and bulk profiling such as TAPIN-seq are complementary approaches. Indeed, we mention this repeatedly in the text, as well as titling Figure 5 to reflect this idea: “TAPIN-seq complements single cell RNA seq profiling”.

We also devote extensive discussion to this theme, including the reviewers suggestion of combining single cell sequencing and TAPIN profiling.

“TAPIN-seq complements single-cell RNA-seq studies of neurons in several ways”“

“Having both deep bulk transcriptomes and single cell maps of the same tissue also provides an opportunity for developing new analytical tools that can harness available cell type-identified information while clustering single cell data.”

“Second, combining our approach with single-cell profiling could more efficiently profile heterogeneity within a brain region or genetically defined cell population”

“Finally, the complementarity between bulk and single-cell measurements extends to other genomic features that can be measured in TAPIN-seq purified nuclei, including accessible chromatin and modified histones.”

In the recent Zipursky lab single cell study of T4/T5 neurons (see above), a GAL4 driver was used to mark cells for presorting by FACS prior to scRNASeq. In this case, the results provide strong support for the notion that anatomical features can be reliable criteria for cell type classification (both anatomy and single cell sequencing indicate exactly eight T4/T5 subtypes).

We anticipate that similar experiments with other optic lobe neurons using our driver lines would produce similar results (i.e. support for anatomically and genetically defined cell types), but do not exclude the possibility that there could be unexpected subdivisions within some of the cell types we profile (as we now discuss).

The reviewer suggests that unbiased clustering of single cell data is somehow more informative (“uses multiple principal components”, a dimension reduction of the 15K gene abundances) and better powered to detect cellular differences than TAPIN-seq profiles of cells that “share one or two enhancers” (although we also measure 15K gene levels). We think this might reflect a mis-understanding of what exactly the one or two enhancers represent. The key feature of our driver lines is not the number of elements used in their construction but the specificity of the expression patterns. The cells labeled by one of these GAL4 or split-GAL4 lines are not randomly related, but through extensive anatomical efforts, are believed to be of the same (anatomically-defined) cell type (or group of cell types). Therefore, the expectation, and our observation in this paper, are that the transcriptomes we measure in these cells using TAPIN-seq in fact reflect the transcriptome of specific cell types.

Bulk and single-cell data are difficult to compare for a number of reasons, that we discuss in the manuscript. Within-cell heterogeneity could potentially be one reason, but we doubt that is the main reason. In fact, as we showed, the unbiased clustering of single cell maps often merge cells of different types, which TAPIN-seq is able to resolve thanks to the genetic tools that differentially mark them. Even if the single cells were correctly clustered so that each cluster does reflect a cell type, their transcriptomes would still be difficult to compare directly to bulk transcriptomes, as the measurement characteristics are quite different, with single cell data using current technologies exhibiting high levels of dropout, as we describe in the text. Nevertheless, as we showed, comparing the two datasets is still a fruitful way of overcoming the main limitations of both: the requirement for genetic tools of TAPIN-seq, and the difficulty of resolving and labeling cell types in scRNA-seq. We also further emphasized the potential of combining bulk and single cell methods in the Discussion:

”Second, combining our approach with single-cell profiling could more efficiently profile heterogeneity within a brain region or genetically defined cell population. Such a combined approach could also help to characterize known cell types for which specific driver lines are not yet available or help to identify and exclude contributions from “off-target” cells to a bulk profile.”

The lack of support for a mid-level hierarchical clustering in the TAPIN-seq profiles is not contradicted by a discrete clustering of single cell data. Whether single cells of a type cluster together is not the same as asking which other cell types/clusters they are most closely related to. We went to lengths to evaluate the uncertainty within our clustering trees and present the confidence values alongside the tree. In contrast, the trees presented in most single cell approaches, either do not perform this evaluation or do not present it alongside the tree. For example, the tree presented in Figure 3 of Konstantinides et al., 2018 does not show this type of evaluation. They do show a similar bootstrapped evaluation as a supplemental Figure (S4), and this shows that very few of their branch points have significant statistical support. In brief, just because you can build a tree doesn’t mean it accurately reflects the transcriptome measurements, let alone provide reliable clues to biological features such as developmental or evolutionary lineage that might conceivably be reflected in such trees.

– Although this is a resource paper,the resource is not accessible to people with limited programming or computational skills. In the web portal opticlobe.com, a simple heatmap can be generated based on the user input. This visualization can hardly reflect the depth and complexity of the data. Please refer to https://gtexportal.org/home/ as a reference to add more tools for general users.

We agree with the reviewer that access to the data is key. For this reason, we have made the expression tables available for download so that any user can explore them using user-friendly spreadsheet programs, such as Microsoft Excel. To remove the hurdle of downloading and opening the table in Excel, we also developed a web portal to support simple queries. We see this web portal mainly as a tool for initial queries which can then be followed up by downloading the spreadsheets or even reanalyzing the original data. This web portal has been useful for others in the field (eg, doi 10.1523/JNEUROSCI.3213-18.2019).

We agree that a better developed web interface would be ideal, and the GTEX portal is indeed an informative guide for this: This project received $800K from the NIH this year to develop “a portal and integrative collaborative analysis platform for GTEX” (https://projectreporter.nih.gov/project_info_description.cfm?aid=9729842&icde=47359899). This kind of effort is beyond the scope of this paper.

– The description of the mixture model is confusing. The lowercase and uppercase p is mixed for different probabilistic events. Do they denote different functions? Conditioned on "bimodal", the posterior probability equation in subsection “Inferring expression state from transcript abundance” is very confusing. Did you integrate out the p(bimodal)? If not, how is the Bayesian rule applied here? The math equations are not well defined.

We thank the reviewer for bringing the confusion about lowercase and uppercase p to our attention. In the statistics literature, uppercase P refers to the probability of a discrete event and lowercase p refers to a continuous density function. We fixed three cases where we inconsistently used uppercase P where it should have been lowercase p.

Yes, in a discrete sense, we did integrate out p(bimodal). As we described in the text, we used the PSIS-LOO approach to selecting whether each gene was better described by a unimodal or bimodal model. Depending on this binary call (either unimodal or bimodal was selected), we chose either the p(expressed | bimodal) or the p(expressed | unimodal). This is equivalent to using the model selection to decide p(gene type = bimodal) = 1 or p(gene type = unimodal) = 1.

– On the model selection section, the first equation used sums of probability, the second equation uses a log sum of probability. Is there a reference or proof for such modeling choices from STAT or machine learning literature?

This math is widely adopted in the Bayesian inference field, and comes from the PSIS-LOO paper that we cite at the beginning of this section (See Equations 20 and 21 from Vehtari et al., 2015; https://arxiv.org/pdf/1507.04544.pdf).

– The mixture model is well studied in both Bayesian and Frequentist framework, how does this approach compare to the standard Bayesian Gaussian mixture model?

Our model is mostly a standard bayesian mixture model, with the wrinkle of the model selection process that we describe in the paper. It would have been ideal if there was a single hierarchical model that jointly learned both the gene type (bimodal vs unimodal) as well as the gene-specific on/off distributions. However, we were unable to build such a model, and the one that we did ultimately build was sufficient to help us successfully navigate our transcriptome catalog.

– Figure 3C is the main result of the mixture model and the lower heatmap showed about HALF of the cells in the matrix are read and the other half is blue across all the genes and samples. This is likely due to an artifact of the customized mixture modeling. Since most of the results in the manuscript depend on the probability estimated from the mixture model, it is crucial to show mathematically that the model is correct.

While there are indeed several choices that go into the modeling, we go through extensive comparison to published expression datasets and extensive protein-level validation of two genes across thirty cell types. In our hands, the probability model was a useful guide for navigating our data. In the case that one is not comfortable with the model calls, they can also use the raw expression levels, which we also provide for download.

In reality, all mathematical models are wrong. As we described in the text, genes are not just on and off; different RNA species have different degradation rates, different translation rates, etc. Like all experiments, every step in a transcriptome study suffers from various kinds of artifacts, ranging from the raw measurements to interpretation.

But in practice, we showed that the measurements are informative, recapitulate known biology, serve as a guide to interpret other genomics datasets, are consistent and corroborated by connectomics data, and provide new insight into the *Drosophila* visual system.